

# An Analysis of Cloud Microphysical Features over UAE Using Multiple Data Sources

Zhenhai Zhang[1], Vesta Afzali Gorooh[1], Duncan Axisa[1], Chandrasekar Radhakrishnan[2],
Eun Yeol Kim[2], Venkatachalam Chandrasekar[2], Luca Delle Monache[1]

[1.] *Center for Western Weather and Water Extremes, Scripps Institution of Oceanography, University of California San Diego, La Jolla, CA, USA*

[2.] *Colorado State University, Fort Collins, CO, USA*

Corresponding author: Zhenhai Zhang (zhz422@ucsd.edu)



**Abstract.** Water is a precious resource and is important for human health, agriculture, industry, and the environment. When water is in short supply, monitoring and predicting the current and future occurrence of precipitation-producing clouds is essential. In this study, we investigate the cloud microphysical features in several convective cloud systems in the United Arab Emirates (UAE) using multiple data sources, including aircraft measurements, satellite observations, weather radar observations, and reanalysis data. The aircraft observation dataset is from an airborne research campaign conducted in August 2019 in the UAE. The cloud cases were identified through analysis of cloud spectrometers mounted on the aircraft. Then, we investigated the microphysical features of those cloud cases with a focus on precipitation microphysics. The effective radius of the cloud particles retrieved from geostationary satellite data was compared with the aircraft in-situ measurement. Using the effective radius retrieved from satellite data, we developed a framework to identify five microphysical zones: diffusional droplet growth zone, droplet coalescence growth zone, supercooled water zone, mixed phase zone, and glaciated zone. The identified zones were verified using the aircraft observations, and the transferability of the 5-zone concept was tested using additional cloud cases. The results show that our 5-zone concept successfully detects the microphysical features related to precipitation using satellite data in the UAE. This study provides scientific support to the development of an applicable framework to examine cloud precipitation processes and detect suitable cloud features that could be tracked for further precipitation analysis and nowcasting.



## 1. Introduction

Water remains a vital resource globally, with its significance heightened by climate change and an increased frequency of extreme weather events. The availability and sustainability of water resources affect every sector (e.g., Pimentel et al., 2004), particularly in arid or semi-arid regions (Wehbe et al. 2021). In the United Arab Emirates (UAE), the absence of precipitation in the context of growing population over this region in recent years raises concerns

about food security and coastline resilience (e.g., Murad et al., 2007). Understanding the physical processes of clouds that trigger precipitation formation is critical for sustainable water management and effective preparation for potential water-related risks.

Most regions in the UAE are arid or semi-arid, and its surrounding areas, except to the north (bordered by the Gulf), are tropical and subtropical deserts (Niranjan Kumar and Ouarda,

2014). The UAE has four climate zones: the Desert Foreland, East Coast, Gravel Plains, and Mountains (Sherif et al., 2014). Most of these zones are characterized by scarce rainfall and a high evaporation rate, except for certain coastal regions. Rainfall distribution within the UAE exhibits large spatial and temporal variation, with the maximum and minimum precipitation occurring in the Mountains and East Coast and the Desert Foreland, respectively (Wehbe et al.

2017). The latter covers the largest portion of the UAE area. The wet season generally occurs from November to April. The average monthly rainfall received by the entire country ranges from approximately 2 mm (e.g., in June) to 15 mm (e.g., in March) (Hussein et al., 2021). The average annual rainfall is generally less than 100 mm, varying from 60 mm to 140 mm (Ouarda et al., 2014; Wehbe et al. 2020). Half of the annual precipitation can fall in a single day during

mesoscale convective events (Wehbe et al. 2019; Kumar and Suzuki, 2019).

Due to the extremely low occurrence of rainfall and dry climate, rainfall enhancement is one of the active areas of research in the UAE (Wehbe et al., 2023). While new technologies for weather modification can improve the operational efficiency of rainfall enhancement activities, identifying suitable targets is always a priority (Axisa and DeFelice, 2016, DeFelice and Axisa,

2016, DeFelice et al., 2023, Hirst et al., 2023). Therefore, it is essential to monitor and detect current and future cloud microphysical features related to precipitation processes.

Within convective clouds, precipitation particles are produced by small-scale microphysical processes that are active in different parts of the cloud. These processes initiate



precipitation through multiple physical pathways at different rates. The efficiency by which
clouds produce precipitation varies greatly and is a function of the dominant physical process
under a specific thermodynamic condition. Growth of precipitation particles can either occur
through collision and coalescence of the ice multiplication process or a combination of the two.
Raindrops cannot form by diffusional growth alone in convective clouds with bases warmer than
0°C. The growth of cloud droplets from a radius of 10 to 20 μm to raindrop size (> 100 μm)
requires an active collision-coalescence process (Bartlett, 1966). In the absence of collision-
coalescence, droplets that form by diffusional growth remain small (i.e., radii < 15 μm) and their
size distribution is composed of a high number concentration of small droplets (Pruppacher and
Klett, 1998). When these droplets reach temperatures colder than 0°C, they become supercooled
and ice can develop through different microphysical pathways. Ice multiplication activity within
supercooled clouds is active in the -5° to -8°C region (Hallet and Mossop, 1974) and the rate of
production of ice depends not only upon the concentration of large drops (> 24 μm diameter) but
also upon the concentration of small drops (< 13 μm) in the cloud (Mossop, 1978). Therefore,
the cloud droplet size distribution and parameters derived from it, along with the cloud
temperature, are critical to understanding cloud microphysical processes and the dominant
physical pathways that lead to precipitation.

Advancement of in-situ and remote sensing technology has provided the cloud physics
community with much improved research tools to study aerosol-cloud-precipitation interactions.
A combination of satellite cloud top temperature and effective droplet radii, retrieved from the
Advanced Very High Resolution Radiometer (AVHRR), has been used to infer the suppression
of coalescence and precipitation processes by smoke (Rosenfeld and Lensky, 1998) and desert
dust (Rosenfeld et al., 2001). Tropical Rainfall Measuring Mission (TRMM) multi-sensor
satellite observations have been used to detect the presence of non-precipitating supercooled
liquid water near cloud tops associated with the heavy seeding from smoke over Indonesia
(Rosenfeld, 1999) and urban pollution over Australia (Rosenfeld, 2000). The time series of
precipitation formation processes within convective storms over the eastern Mediterranean were
tracked by METEOSAT Second Generation (MSG) to investigate the cloud response to aerosol
loading. A strong correlation was found between the aerosol loading and the depth above cloud
base required for the onset of precipitation (Lensky and Shiff, 2007). Aircraft in-situ
measurements of continental convective clouds seeded with finely milled salt powder detected a



broadening of the cloud drop size distribution (Rosenfeld et al., 2010), indicating an acceleration
      of the warm rain process. In addition, aircraft measurements have provided evidence that dust
      particles extend the tail of the cloud droplet size distribution spectra, increasing the droplet
      effective radii and triggering the formation of warm rain (Pósfai et al., 2013).

            Rosenfeld and Lensky (1998) used the AVHRR satellite data to analyze vertical profiles
of the cloud particles' effective radius to investigate the precipitation formation processes in
      convective clouds and introduced five distinct vertical cloud zones, including (1) diffusional
      droplet growth zone, (2) droplet coalescence growth zone, (3) rainout zone, (4) mixed-phase
      zone, and (5) glaciated zone, which characterize the microphysical features of the cloud from the
      precipitation formation perspective. Lensky and Drori (2007) followed the Rosenfeld and Lensky
(1998) approach and defined the temperature of precipitation onset as the temperature where the
      median effective radius exceeds a threshold of 15 μm. A recent study by Wang et al. (2019)
      focused on identifying supercooled water clouds and developed a method to detect them based
      on cloud phase, effective radius, optical thickness, and cloud top temperature from the Advanced
      Himawari Imager and aircraft in-situ cloud measurements.

130         During the last two decades, there has been a continuous effort focused on rainfall
      enhancement science in the UAE (Mazroui and Farrah, 2017; Al Hosari et al., 2021). However,
      the recent enhanced observations, including the airborne measurements over the UAE (Wehbe et
      al., 2021; Wehbe et al., 2023) and remote sensing from geostationary satellites (Meteosat-10 and
      Meteosat-8; Kumar and Suzuki 2019), provide unique data sources to examine the cloud
microphysical features and the dominant physical pathways that lead to precipitation in the UAE.
      Kumar and Suzuki (2019) evaluate the spatial and seasonal occurrence of cloud cover from
      Meteosat-10 and Meteosat-8 in the UAE, and analyze the cloud phase distribution to determine
      the potential for precipitation enhancement through cloud seeding with aerosols. In general,
      cloud seeding is applied using aerosol that is active as a cloud condensation nuclei (CCN) in the
warm part of the cloud (called 'hygroscopic seeding'; Mather et al.,1997; Cooper et al.,1997;
      Bruintjes, 1999; Terblanche et al., 2000; Silverman, 2000; Silverman, 2003; Rosenfeld et al.,
      2010; Flossmann et al., 2019), and aerosol that is active as an ice nucleating particle (INP)
      around supercooled liquid water clouds (called 'glaciogenic seeding'; Bruintjes, 1999;
      Silverman, 2001; Woodley et al., 2003a, 2003b; Flossmann et al., 2019). In both hygroscopic
and glaciogenic seeding, seeding material must be properly applied to be effective (Geresdi et





al., 2021). This is often referred to as 'targeting'. In practice, this is often the most challenging part of operational seeding programs.

In this study, we investigated the microphysical features of cloud cases over the UAE using multiple data sources focusing on the cloud microphysics of precipitation. We examined these features using aircraft observations, introduced a new 5-zone framework to detect the cloud microphysical zones using satellite data, and used aircraft measurements to validate the detected cloud zones. The corresponding synoptic conditions and the radar reflectivity features for cloud cases were also explored. This study aims to develop an applicable framework that detects cloud features that correspond to microphysical pathways that are active in different parts of the cloud, to characterize precipitation processes in the UAE. One application of this framework is in the development of a tool for further analysis of precipitation and nowcasting, and to assist with cloud targeting in operational seeding programs.

## 2. Dataset and methodology

### 2.1 Aircraft observations

The aircraft data is from the UAE 2019 Airborne Campaign (Wehbe et al., 2021; Morrison et al., 2022). The scientific flights in this campaign were conducted by the Stratton Park Engineering Company (SPEC) Learjet 35A in August 2019. This SPEC Learjet 35A aircraft was equipped with state-of-the-art cloud physics instruments. The list of instruments include the following: cloud particle imager (CPI; Lawson et al. 2001); two-dimensional stereo (2D-S) probe (Lawson et al. 2006); high-volume precipitation spectrometer (HVPS; Lawson et al. 1998); fast forward-scattering spectrometer probe (FFSSP; Brenguier et al. 1998); fast cloud droplet probe (FCDP; Lawson et al. 2017; Wood et al., 2018); and Nevzorov hot-wire probe (Korolev et al. 1998). The FCDP, FFSSP, 2D-S, and HVPS were all equipped with probe tips to reduce the effects of ice crystals shattering (Korolev et al. 2011; Lawson 2011) and data were postprocessed using an interarrival time algorithm to remove shattered particles (Lawson 2011). When combined the cloud particle probes can measure size distributions in the range 2 μm to 2 cm diameter. In this campaign, there were 11 scientific flights (SF) listed in Table 1. Figure 1 shows some examples of the flight tracks, including SF01 on August 12[th], SF02 on August 13[th], and SF03 on August 18[th], 2019. Wehbe et al. (2021) study the evolution of growing convective cloud



tops during this campaign, and present aerosol and cloud microphysical measurements from
SF01 and SF04. Morrison et al. (2022) examine microphysical processes with a focus on
studying activation and growth of cloud droplets in a bin model, and comparing modeled droplet
size distributions with observations for case SF01. In this study we focus on the microphysical
properties of SF03 with a focus on the evolution of the drop size distribution and precipitation
microphysics.

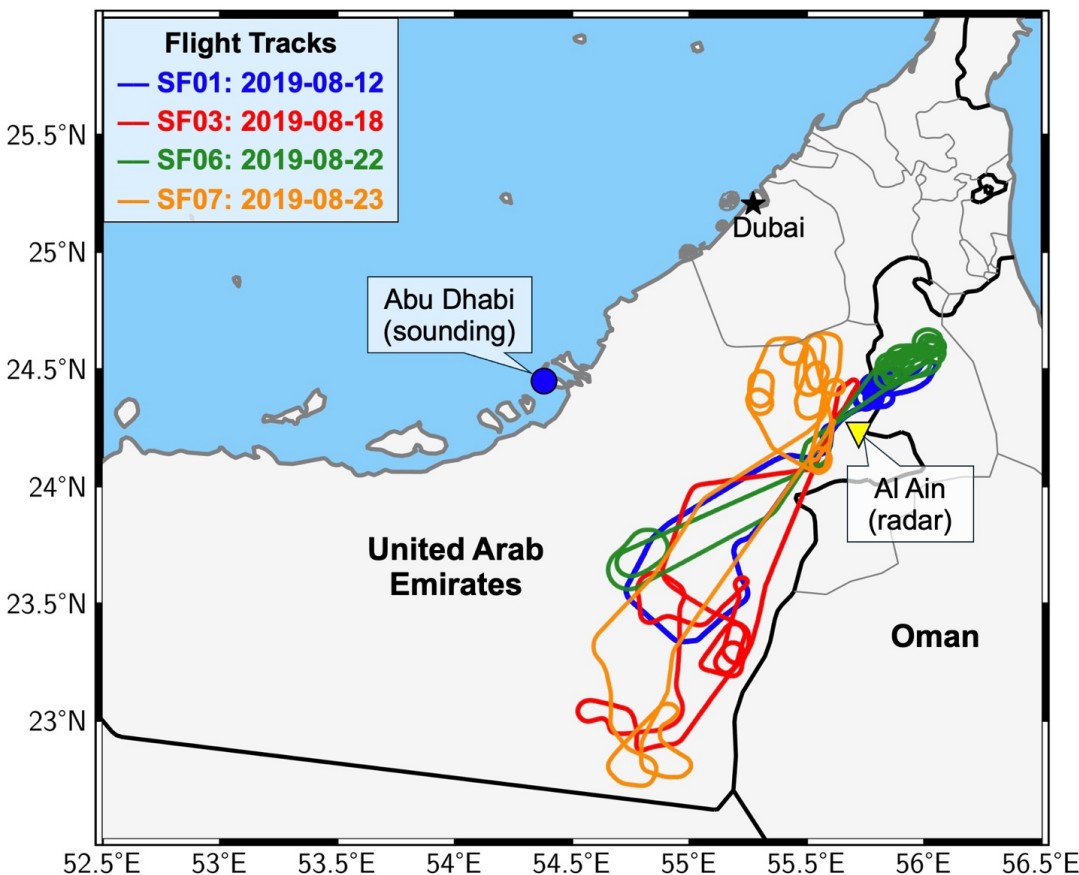

Figure 1. A map showing the flight tracks for SF01 (blue line), SF03 (red line), SF06 (green
line), and SF07 (orange line) of the Lawson Airborne Campaign 2019. The yellow triangle is the
location of the weather radar in Al Ain and the blue circle is the location of the Abu Dhabi
airport for sounding observation.



For a better utilization of the aircraft observation, we first identified the measurements of
the cloud for each cloud penetration by the aircraft. Figure 2a-b is a time segment of the
observed total water content from the Nevzorov hot-wire probe (LWTA) and cloud droplet
concentrations from FCDP (ConFCDP) in SF03. When the aircraft penetrated a cloud the LWTA
and ConFCDP rapidly increased, as highlighted in the dashed boxes. In this study, if the LWTA
is at least 0.05 g m$^{-3}$ and ConFCDP is at least 20 cm$^{-3}$ for one second or longer, it is considered

one cloud penetration (CP). Using this definition, we identified the CPs in all 11 flights, and the
numbers of CPs in each flight are listed in the third column of Table 1. Seven of the eleven
flights have at least 5 CPs. Several sensitivity tests were conducted to examine the impacts of the
thresholds on the defining CPs. Different thresholds in the minimum LWTA (0.01 g m$^{-3}$ and 0.1
g m$^{-3}$) and ConFCDP (10 cm$^{-3}$ and 30 cm$^{-3}$) did not have any significant impacts on the number

of detected CPs since the values of those two parameters in CPs are usually substantially higher
than the minimum thresholds (e.g., Figure 2a-b and Figure 5).

After identifying the CPs, we compared selected parameters from different instruments.
Figure 2c shows a comparison of liquid water content (LWC) from the FCDP and FFSSP probes
with the LWC measured by the Nevzorov probe for the identified CPs in SF03. The Nevzorov

probe is a constant-temperature and hot-wire probe designed for measuring the cloud ice and
liquid water content, which can provide a relatively more accurate measurement of the water
content (Korolev et al., 1998). The LWC from FCDP has a better agreement with the Nevzorov
LWC, and their correlation is 0.91, significantly higher than the correlations between the FFSSP
and the Nevzorov probes (0.67). Thus, in this study, we used the measured parameters from

FCDP, including cloud particle size distribution and its derived parameter cloud droplet effective
radius.







Figure 2. Panels (a) and (b) show examples of cloud penetrations for flight segments when the
aircraft penetrated clouds. (a) Time series for total water content (LWTA) and the minimum
LWTA threshold (blue dashed line) for cloud penetrations. (b) Time series for cloud particle
concentration from FCDP and the minimum FCDP concentration threshold (red dashed line) for
cloud penetrations. (c) Comparison of liquid water content from different instruments.



A vertical distribution of the mean effective radius for all the identified CPs in the flights
        is shown in Figure 3. The SF03 has a cloud base at 9.0°C, around 3.5 km of elevation, and the
        highest/coldest CP is at about -13.0°C, near 6.9 km, indicating a relatively deep cloud. It is worth
        noting that the cloud top could be higher than the highest CP measured by the aircraft and other
        colder cloud tops could have been present. The cloud droplet effective radius (ER) is about 4.8

μm at the cloud base and increases with height (decrease of temperature) with a maximum ER of
        8.9 μm at -8.2°C and -12.9°C. While many cases have a cloud base temperature similar to that of
        SF03 at around 8.0°C to 9.0°C, SF07 and SF06 have a relatively high measured cloud base with
        temperatures at 1.8°C and -3.3°C, respectively. All these cloud cases were analyzed with aircraft
        observations, and SF03 was utilized as a prime example to demonstrate our analysis using

aircraft observations in Section 3.

| UAE 2019 Airborne Campaign | | | | |
|---|---|---|---|---|
| **Flights** | **Date** | **Cloud Penetration #** | **Temperature (°C)** | **Mean Effective Radius (μm)** |
| SF01 | 2019-08-12 | 38 | -13.6 – 9.3 | 4.6 – 9.0 |
| SF02 | 2019-08-13 | 45 | -11.7 – 8.0 | 3.5 – 14.3 |
| SF03 | 2019-08-18 | 29 | -13.0 – 9.3 | 4.4 – 8.9 |
| SF04 | 2019-08-19 | 58 | -13.1 – 9.3 | 3.1 – 8.7 |
| SF05 | 2019-08-20 | 0 | / | / |
| SF06 | 2019-08-22 | 15 | -16.1 – -3.2 | 4.4 – 9.9 |
| SF07 | 2019-08-23 | 66 | -12.0 – 1.8 | 3.4 – 8.0 |
| SF08 | 2019-08-24 | 5 | -9.6 – -8.6 | 7.8 – 8.4 |
| SF09A | 2019-08-26 | 0 | / | / |
| SF09B | 2019-08-26 | 0 | / | / |
| SF10 | 2019-08-28 | 0 | / | / |

Table 1. A summary of the 11 scientific flights (SFs) in the Lawson Airborne Campaign 2019, including the flight date, number of identified cloud penetrations (CPs), the cloud temperature range of the CPs, and the mean effective radius range of the CPs.






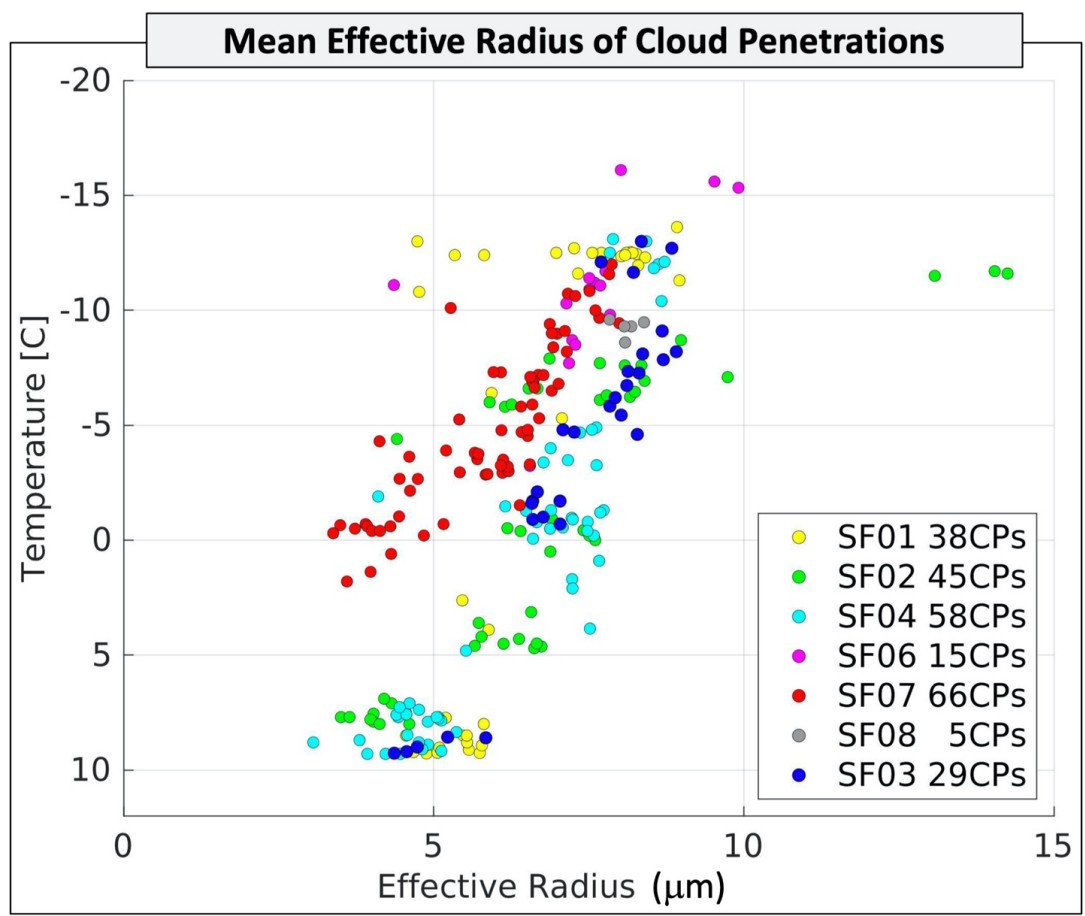

Figure 3. The distribution of the mean effective radius of each cloud penetration. Different colors indicate different scientific flights (SF).

## 2.2 Satellite products

*(a) High Rate SEVIRI Level 1.5 Image Data*

Our study uses the near real-time high spatiotemporal resolution data from the Spinning Enhanced Visible and Infrared Imager (SEVIRI) sensor onboard the METEOSAT Second Generation (MSG) satellite (Meteosat-8) with Indian Ocean Data Coverage (IODC). The Visible (VIS), Near Infrared (NIR), and Infrared (IR) bands are geolocated using a standardized projection, resulting in images containing calibrated, radiance linearized, and Earth-located information, which is appropriate for deriving meteorological products and conducting additional



meteorological processing. We use the derived cloud brightness temperature (T) from one IR

channel with a center wavelength of 10.8 μm known as a "clean" longwave IR window. This

channel is characterized by lower sensitivity to water vapor absorption, enhancing atmospheric

moisture correction, and facilitating continuous day/night cloud and convection feature

identification. The nominal IR image sampling distance (i.e., spatial resolution) is 3 km by 3 km

at the sub-satellite point, and the temporal resolution is 15 mins (96 data points per day).

*(b) Optimal Cloud Analysis*

This study includes the use of the upper layer cloud effective radius (ER) and cloud phase

retrievals from SEVIRI Optimal Cloud Analysis (OCA) algorithm. The OCA method uses

reflectance from VIS and NIR channels, radiances from IR channels, the European Centre for

Medium-range Weather Forecasts (ECMWF) forecast variables, surface reflectance maps as well

as cloud mask products to provide ER ranging from 1 to 31 μm. The SEVIRI OCA scheme is

based on an optimal estimation (OE; King et al., 1997; Watts et al., 2011), and it is beneficial for

convective cloud monitoring over the Middle East (Mecikalski et al., 2011; Larzi et al., 2014;

Hadizadeh et al., 2019). These products rely on the principle that the cloud's optical thickness

predominantly determines the reflection function of clouds at a non-absorbing band. In contrast,

the reflection function at a water (or ice) absorbing band mainly depends on the size of cloud

particles.

Originally, OCA was conceived as a research endeavor at the Rutherford Appleton

Laboratory (RAL) in 1997, the OCA product has since evolved into an operational tool

developed by EUMETSAT to deliver timely cloud parameter retrievals from the MSG SEVIRI

instrument (Watts et al. 1998). Notably, the OCA product distinguishes itself from alternative

retrieval methods by relying on a comprehensive cost function value, indicating consistency

between modeling and reality. While challenges persist, including nighttime performance

limitations and constraints in detecting multi-layer conditions for moderate cirrus optical depths,

the OCA algorithm remains a cornerstone in advancing our understanding of cloud dynamics and

their impacts across various scientific domains. OCA approaches cloud retrieval as an inverse

problem, utilizing a forward model using a radiative transfer model (RTM) to simulate satellite

radiances based on a parametrized cloud/atmosphere/surface model and defined observing



conditions. The OE method is then employed to obtain cloud parameters that best match observed radiances, considering measurement errors and prior knowledge (Rodgers, 2000). The
OE maximizes the probability of the retrieved state (e.g., cloud effective radius) conditional on the value of the measurements and any a priori knowledge (Poulsen et al., 2012; Watts et al., 2011).

This iterative process aims to minimize a cost function by adjusting the state vector, utilizing the Levenberg-Marquardt scheme for optimization. To initiate the minimization
process, the model begins with an initial guess state, typically set to the value of the a priori without additional information. Subsequently, it proceeds by iteratively adjusting the state vector in the direction that decreases the cost function at each step. This iterative approach ensures that the updated vector progressively converges towards the minimum of the cost function. Convergence is reached when the cost function changes minimally between iterations, with
unreached convergence deemed as invalid retrievals. The value of the cost function serves as a measure of the solution-state's consistency with observations and prior knowledge, with high or low values indicating potential overestimation or underestimation of error, respectively. It is assumed that measurement errors, a priori parameters, and the forward model follow a Gaussian distribution with a zero mean and covariances.

Phase determination is a crucial aspect of cloud property retrieval, although it is not directly included in the state vector due to its binary nature. In the EUMETSAT OCA approach, the cloud phase is initially assumed to be either ice or liquid based on the calculated overcast brightness temperature of the 11 μm channel, with a threshold of 260 K distinguishing between the two (Mixed phase is not explicitly accounted for in the OCA approach). Throughout the
retrieval iteration, the phase may be switched based on specific criteria: a change from liquid to ice occurs when the estimated effective radius exceeds 23 μm, prompting a restart of the retrieval assuming ice phase; conversely, if the effective radius for ice cloud falls below 20 μm, the retrieval is restarted assuming liquid phase. In our study, we consider the clouds as ice (hereafter ice cloud), liquid (hereafter water cloud), and total cloud (without any classification).


### 2.3 Other datasets



There are six C-band weather radars covering the UAE region. In this study, the dual-polarization vertical profiles from Al Ain radar (Figure 1) were used to analyze cloud characteristics because the observation area of that radar has overlaps with the aircraft

observation. The quality control (QC) procedure was performed before generating the radar's dual-polarization vertical profiles. The QC procedure includes detecting and removing Radio Frequency Interference (RFI), sea clutter, and noise from the data. RFI and sea clutter removal are based on a fuzzy logic algorithm (Liu et al., 2000), and the censoring of noisy data is performed using Signal-to-Noise Ratio (SNR) and Normalized Coherent Power (NCP). After the

QC process, the radar data were converted from polar to Cartesian coordinates, with both horizontal and vertical resolutions set at 0.5 km. In the final step, using the location and time of the cloud penetrations from aircraft observation we identified coincident observation from the gridded radar data. We applied a 1 km spatial and a 5-minute temporal threshold in the collocation procedure. The radar vertical profiles are generated from the coincident observations.

The latest version of the reanalysis dataset from the European Centre for Medium-Range Weather Forecasts (ERA5; Hersbach et al., 2020) was used in this study to provide an overview of weather conditions for the cloud cases, including the total cloud cover, total column water, and Convective Available Potential Energy (CAPE). The ERA5 data is on a horizontal resolution of about 31 km and 137 vertical levels from the surface up to 0.01 hPa (~80 km). The data used

in this study is obtained on 0.25-degree horizontal resolution and hourly temporal resolution. In addition, the temperature sounding profile observed in the Abu Dhabi airport (Figure 1) is used to explore the temperature inversions.

### 3. 18 August 2019 (SF03) case study

**3.1 Meteorology**

First, a synoptic overview for 18 August 2019 (SF03) was conducted using ERA5 and satellite data (Figure 4) to understand the overall weather conditions, including the total cloud cover, total column water, and Convective Available Potential Energy (CAPE) at 13 UTC from ERA5 and the 3-hour precipitation amount from satellite data at 12-15 UTC. Meanwhile, SF03

was conducted from 12:53 –- 14:31 UTC, and all the CPs were located within the small black box in Figure 4. The atmosphere within this observation area (the small black box) contained a



substantial amount of water vapor, measured as 40–45 mm total column water (Figure 4a). The CPs were located on the east side of a strong convection zone (Figure 4b), and the total cloud cover was 40–70% (Figure 4c). Meanwhile, there is a strong temperature inversion layer around

6 km altitude at a temperature of -5°C – -7°C over this area according to the temperature sounding profile observed in the Abu Dhabi airport at 12 UTC and the ERA5 reanalysis (not shown). Temperature inversions are frequently observed during summer in UAE (Weston et al., 2020), which can suppress convection. Based on the satellite precipitation amount during 12-15 UTC, the SF03 CPs were over southeast of a precipitation area (Figure 4d). In this study, we

only used the reanalysis to explore the synoptic conditions for the cloud case. To identify potential cloud targets for rainfall enhancement applications, high-resolution short-term numerical weather forecasts or nowcasts provide useful information (total column water, CAPE, total cloud cover, etc.) to locate potential areas for convective cloud development.



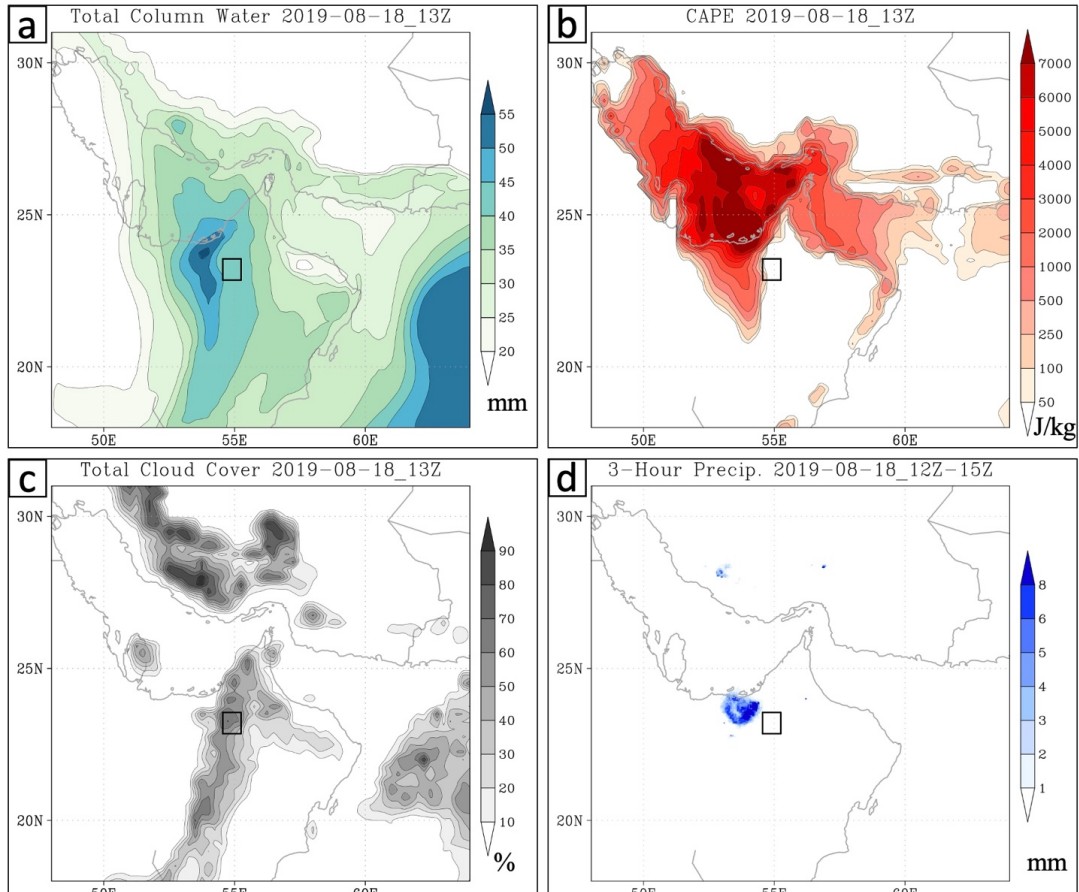

Figure 4. (a) the total column water (mm) at 13 UTC on August 18[th], 2019, from ERA5 reanalysis; (b) same as (a) but for the Convective Available Potential Energy (CAPE, J/kg); (c) same as (a) but for the total cloud cover (%); (d) the precipitation amount (mm) during 12-15 UTC on August 18[th], 2019, from satellite data. The small black box in each panel shows the location of cloud penetrations identified in SF03.

## 3.2 Analysis of cloud microphysical parameters from aircraft observations

Aircraft observations provide detailed measurements of the cloud microphysical properties within clouds. The time series of a few selected parameters from SF03 on August 18[th], 2019, was shown in Figure 5 as an example. This flight took off at 12:53 UTC and landed at 14:31 UTC. The coincidence of the peak values in total water content from the Nevzorov hot-wire probe and FCDP concentrations showed good agreement in detecting CPs. The first CP



occurred at 13:14:34 UTC with observed peaks in both total water content of 1.19 g m$^{-3}$ and FCDP concentration of 860 cm$^{-3}$ at a temperature of -4.6°C and a height of 5.8 km. The highest

(coldest) CP was conducted at 13:26:35 UTC at a height of 6.9 km and a temperature of -13°C with total water content of 0.24 g m$^{-3}$ and FCDP concentration of 48 cm$^{-3}$. At 14:05:34 UTC, CPs were detected at around 3.6 km height with a temperature of around 9.1°C, and below that, no other CPs were detected, which indicates a cloud base of 3.6 km with a total water content of 0.25 g m$^{-3}$ and FCDP concentration of 1286 cm$^{-3}$. In total 29 CPs were identified for this case.

The mean effective radii (see Figure 3) range from 4.6 to 8.9 μm throughout the vertical profile of the cloud and smaller than the 15 μm threshold for onset of warm rain (Lensky and Drori, 2007). Although some CPs occurred very close to each other (a few seconds away), all the CPs were identified and processed as independent CP measurements.





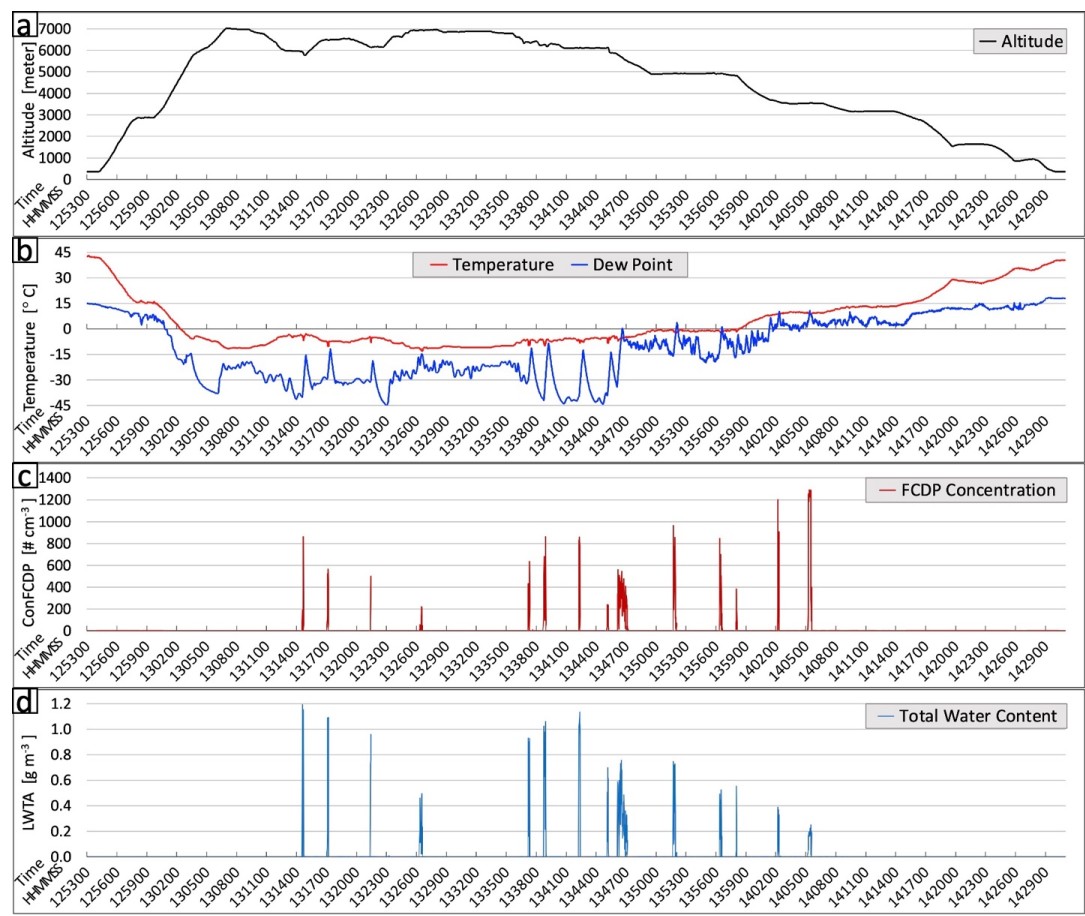


Figure 5. The aircraft observation from SF03, including (a) altitude, (b) temperature and dew point, (c) cloud particle concentration from FCDP, and (d) total water content.

The cloud particle size and the growth rate of the size with height are critical to the

formation of rain-sized droplets (Freud and Rosenfeld, 2012). Overall, the ERs of the CPs from

SF03 increase with the decrease in temperature (left column of Figure 6). Four CPs from the

cloud base to the cloud top are selected for further analysis, including the cloud particle size

distributions (middle column), and 2DS and CPI images (right column) from the research

aircraft. The first CP (first row of Figure 6) is around the cloud base with a temperature of 9.1°C.

Figure 6b shows the corresponding cloud particle size distribution from three instruments:

FCDP, 2DS, and HVPS. The FCDP measures the size and number concentration of cloud

droplets in the range of 2 μm to 50 μm diameter (Lawson et al. 2017). This cloud particle size



distribution for this CP has the highest concentration at the particle diameter 7–9 μm, decreasing
with increasing size (red trace in Figure 6b). The minimum detectable cloud particle size of 2DS
is about 10 μm diameter (Lawson et al., 2006; Baker et al., 2009), and is in good agreement with
the FCDP for cloud particle size larger than 20 μm. Overall, the cloud base penetration with a
temperature of 9.1°C has a high droplet concentration at a relatively small size (< 20 μm). This is
consistent with the 2DS images for this CP, where large droplets are absent in the 2DS image
strips (e.g., Figure 6c). At colder temperatures of -0.7°C (second row of Figure 6), particle size
distribution shifts to larger sizes with a maximum concentration of around 10 μm in the FCDP,
showing very little droplet growth in the main body of the size distribution in the warm part of
the cloud. At a temperature of -5.2°C (third row of Figure 6), the size distribution has a peak
concentration at 11 μm, and the 2DS detects a small number concentration (21 per liter) of
particles at 50-200 μm in diameter. The 2DS images a few particles around 100 μm in diameter,
and the inspection of the CPI shows that these are spherical and around 100 μm in diameter
(Figure 6i). At a temperature of -12.1°C (fourth row of Figure 6), there are significantly more
cloud particles around or larger than 100 μm based on 2DS with a concentration of 137 per liter
(Figure 6k). The 2DS and CPI image particles are ice and larger than 100 μm in diameter (Figure
6l). The observed ice particles indicate an ice production process that is active around -12°C in
case SF03.

        The noticeable features of these observations is that the cloud base temperature, effective
radii, droplet number concentrations and total water content are typical of high based continental
convective clouds that are composed of a high number concentration of small droplets formed by
diffusional growth of droplets. These droplets have low collision efficiency, and the collision-
coalescence process is suppressed as evidenced by the particle size distributions in Figure 6b
where the peak concentration and the effective radius is smaller than the 15 μm threshold for
warm rain (Lensky and Drori, 2007). The cloud penetration at -12°C indicates that ice production
is active at warmer temperatures, which is consistent with ice multiplication within supercooled
clouds in the -5° to -8°C region (Hallet and Mossop, 1974) that depends on the ratio of small
(diameter < 13μm) to large (diameter > 24 μm) cloud droplets.









Figure 6. The first row is for the cloud penetration at 9.1°C: (a) the effective radius of each cloud penetration from SF03 and the penetrations at 9.1°C is highlighted in a red circle; (b) the distribution of cloud particle size; (c) 2DS images (top) and CPI images (bottom). (d-f), (g-i), and (j-l) are the same as (a-c) but for cloud penetrations with temperatures of -0.7°C, -5.2°C, and -12.1°C, respectively.

In general, the analysis of cloud microphysical parameters observed in case SF03 agrees with Wehbe et al. (2021) for SF01 and SF04 where the dominance of small-sized particles with diameters less than 10 μm and the minimal concentrations of intermediate sizes (10-30 μm) indicates that an active collision-coalescence process was not achieved. Wehbe et al. (2021) postulated that strong updrafts in SF04 may have carried a limited number of large particles aloft to serve as INP at -10.6°C, but not in SF01. Although the occurrence of first ice cannot be linked to a specific ice nucleation process, ice production is active in SF03. There are many uncertainties associated with the number concentration of ice particles expected within high based continental convective clouds within a certain time, especially in a dusty boundary layer where INP concentrations in the Arabian Basin range up to 2 orders of magnitude at -15°C, between $5\times10^{-3}$ and $5\times10^{-1}$ L$^{-1}$ (Beall et al. 2022). However, the tail in the particle size distribution larger than 100 μm (see Figure 6k) in SF03 is indicative of an active ice production process that is dominant compared to a suppressed collision-coalescence process (see Figure 6b) where the size distribution shows a high number concentration of small droplets.

### 3.3 Effective radius from satellite data

While aircraft observations can provide detailed measurements to examine the microphysical features of the cloud, it has a limited sample size of measurements and is usually not available for routine assessment of dominant physical pathways that lead to precipitation. Thus, more accessible observation data is needed for real-time applications, such as satellite data. As described in Section 2.2, we retrieved the cloud particle ER for each cloud case using the satellite data from the SEVIRI – METEOSAT 2nd Generation Indian Ocean Dataset.





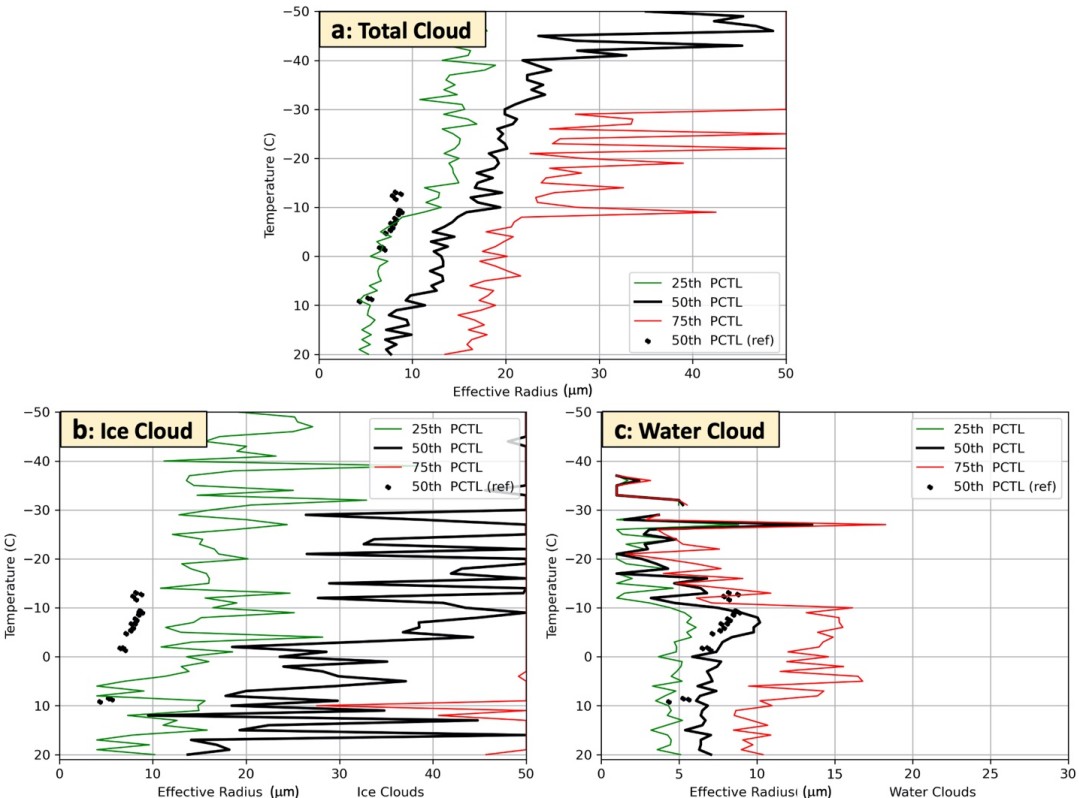

Figure 7. (a) shows the 25th (green line), 50th (black line), and 75th (red line) percentiles of effective radius (ER) for the total cloud in SF03 retrieved from satellite data compared with the ER from aircraft observation (dots, 50th percentile). (b) and (c) are the same as (a), but for the ice and water clouds, respectively.

Before utilizing the ER values retrieved from satellite data, they are validated with the ER measured by aircraft. Figure 7 shows the comparison of ERs between the satellite and aircraft datasets for case SF03. In each CP from the research aircraft, the 25th, 50th, and 75th percentile of the ER values are very close to each other because the ER measured from most aircraft CPs are in a short time (one to several seconds/measurements per CP). Therefore, only the 50th percentile of ER values is plotted as black dots in Figure 7. Compared to the satellite-retrieved ERs for total cloud, the aircraft-measured ERs are around the 25th percentile of the satellite ERs (Figure 7a). This is similar to the results from Rosenfeld and Lensky (1998), who showed that the ERs measured by aircraft were mostly around the 25th percentile of the satellite measurements. The





ERs measured by aircraft are further compared to the ERs of water clouds and ice clouds from the satellite. The results show that the aircraft ERs are close to the 50th percentile of the satellite ERs for water clouds, indicating a good agreement between these two datasets (Figure 7c). That

is because the aircraft ERs are calculated from FCDP particle size distribution, which measures the size of particles in the 2-50 μm diameter range and is sensitive to water droplets. It is not surprising that the aircraft ERs tend to be smaller than the satellite ERs for ice clouds due to the lack of sensitivity of the FCDP to ice particles. Overall, the ERs from aircraft and satellite datasets have a fair agreement, which gives us the confidence to use ERs retrieved from satellite

data to analyze the relevant cloud features.

### 3.4 Definition of the Cloud Zones

When there is sufficient data, the next step is characterizing the cloud microphysical features that are indicators of the dominant microphysical processes leading to precipitation.

Rosenfeld and Lensky (1998) introduced a 5-zone concept for some cloud cases based on their microphysical features, including diffusional droplet growth zone, droplet coalescence growth zone, rainout zone, mixed phase zone, and glaciated zone. In this study, we follow the Rosenfeld and Lensky (1998) concept and propose a refinement to their methodology. To better represent the early development of convective clouds, we replaced the rainout zone with a supercooled

water zone, where supercooled droplets are a hydrometeor type associated with ice production and growth of ice particles to precipitation sizes in mixed-phase convective clouds. In addition, we added the thresholds of brightness temperature, ER, the growth rate of ER, and the cloud phase to define the zones to correspond with basic cloud physics principles described in Section 1. Figure 8 is a framework to detect the 5 zones using satellite data. The thresholds in the

framework were determined based on the analysis of our case SF03, then tested using different cloud cases (SF01, SF06 and SF07) and validated using aircraft observations (Section 4). The definitions of the 5 zones and the corresponding thresholds for satellite data are listed below.

(1) Zone 1, diffusional droplet growth zone: It is usually close to the cloud base with relatively small particle size and very slow growth of size. Thus, it is detected using the satellite data of

the water phase cloud and identified when brightness temperature (T) > 0°C, 50th percentile of ER < 10 μm. When this zone exists and is relatively deep, the microphysical processes



favorable to precipitation initiation are suppressed, indicating potential for rainfall enhancement.

(2) Zone 2, the droplet coalescence growth zone: The particle size's growth rate in this zone is large, indicating a quick cloud particle growth above the cloud base through a collision-coalescence process. Thus, it is detected using the water phase cloud and identified when T is lower than the T in Zone 1 and higher than -10°C, the 75th percentile of ER is between 15 to 20 μm, and the growth rate is relatively large (dER/dT < -0.4 μm per °C). When Zone 2 exists and is deep, the microphysical processes favorable to precipitation initiation are active, and the potential for rainfall enhancement is low.

(3) Zone 3, the supercooled water zone: This zone has water particles at a temperature considerably below the freezing temperature, and the growth rate of the particle size is relatively slow. Thus, it is detected using the water phase cloud and identified when 0°C > T > -38°C, the 50th percentile of ER < 20 μm and the growth rate is between -0.4 – 0.0 μm per °C. When Zone 3 exists and is sufficiently deep, the microphysical processes favorable to precipitation initiation are usually suppressed, indicating potential for rainfall enhancement.

(4) Zone 4, the mixed phase zone: This zone has a mixed phase with relatively large particles and rapid particle size growth that usually occurs at relatively low temperatures. Thus, it is detected using satellite data of the total cloud and identified when -10°C > T > -38°C, the 75th percentile of ER > 20 μm, and the growth rate dER/dT < -0.4 μm per °C. A deep Zone 4 usually indicates suppressed microphysical processes and potential for rainfall enhancement.

(5) Zone 5, the glaciated zone: It is a nearly stable zone of ER, and the glaciated particle size is usually large. Thus, it is detected using the ice phase cloud and identified when T is lower than -10°C and the 75th percentile of ER > 25 μm. If Zone 5 exists, the cloud has active microphysical processes, which indicates potential for rainfall enhancement is low.

Generally, when the cloud is identified as having suppressed microphysical processes, it could be a suitable target for precipitation enhancement. Figure 8 briefly defines the identification of hygroscopic seeding and glaciogenic seeding patches at the bottom-right corner, which will be discussed in the last section Summary and Discussion.



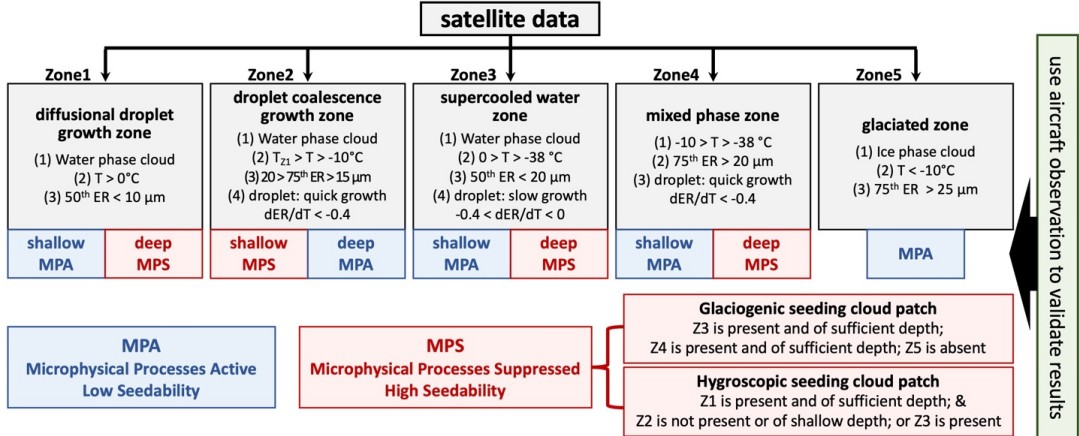

Figure 8. The flowchart of the 5-zone framework, including (1) the diffusional droplet growth zone, (2) the droplet coalescence growth zone, (3) the supercooled water zone, (4) the mixed phase zone, and (5) the glaciated zone. The blue box indicates microphysical processes active (MPA), and the red box indicates microphysical processes suppressed (MPS).

An example of a cloud patch at a specific time from SF03 is utilized to demonstrate how to use the 5-zone framework to identify the zones. Figure 9a shows some cloud patches (different colors) from the satellite data on August 18th, 2019, and the one highlighted in a red circle is selected for detecting the zones. Figures 9c-d show the ERs for the water, ice, and total cloud phases. In the water cloud, the algorithm detects deep Zone 1 (vertical purple bar) and Zone 3 (vertical cyan bar) layers; and Zone 2 is not detected. In the total cloud, the 75th percentile of ER grows quickly between the temperatures of -37°C and -11°C, identified as Zone 4 (vertical yellow bar). Meanwhile, Zone 5 is identified at a temperature lower than -39°C (vertical orange bar) using the ice clouds satellite data. Overall, due to the present and sufficient depth of Zones 1, 3, and 4, this cloud patch is categorized as microphysical processes suppressed (MPS), indicating it is a potential target of cloud seeding.





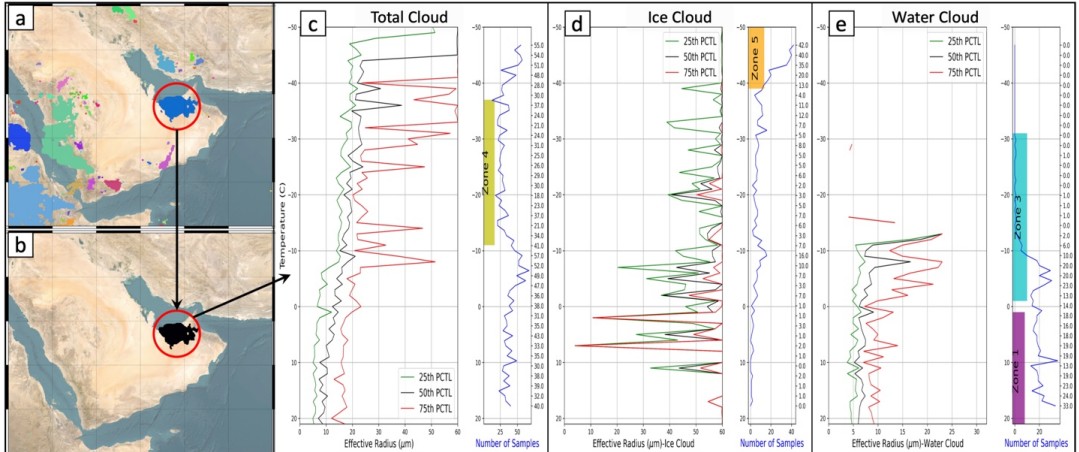

Figure 9. (a) Examples of cloud patches (colors) detected from satellite data. (b) The selected cloud patch for the analysis of effective radius. (c) The effective radius from satellite data for total cloud (left) and the number of data samples (right); the vertical yellow bar represents identified Zone 4, mixed phase zone. (d) is the same as (c) but for the ice cloud and the vertical orange bar represents identified Zone 5, glaciated zone. (e) is the same as (c) but for the water cloud and the vertical purple and cyan bars represent identified Zone 1 (diffusional growth zone) and Zone 3 (supercooled water zone), respectively.

## 4. Transferability of cloud zones to other cloud cases

In the previous section, we focused on the cloud case of SF03 and introduced the 5-zone framework to identify different cloud microphysical zones. In this section, the 5-zone framework is tested in more cloud cases, and the results are validated using aircraft observation to evaluate the transferability of the 5-zone concept.





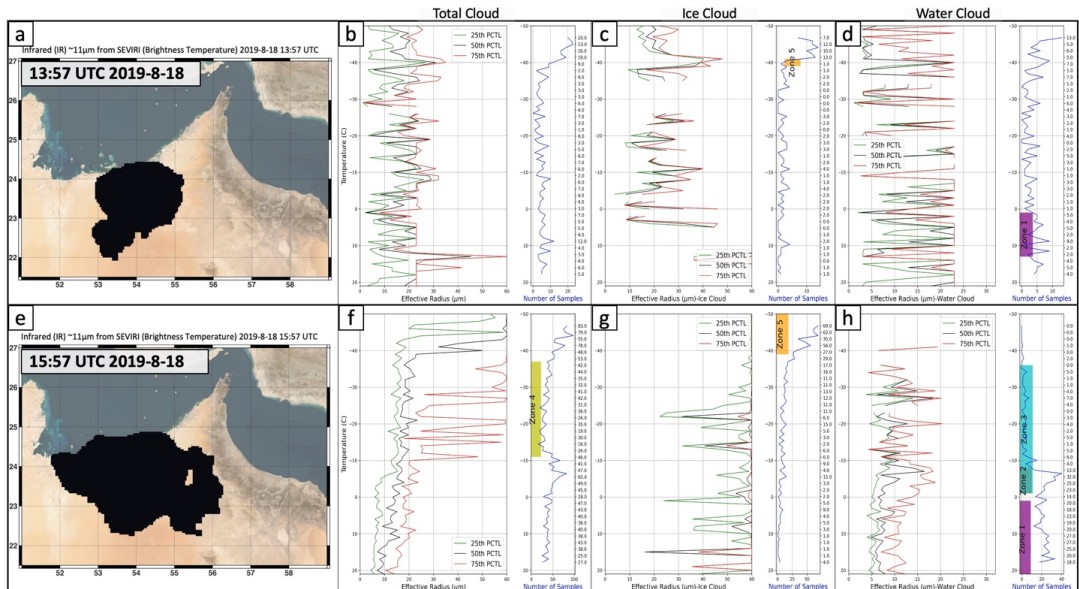

Figure 10. The detected zones in one cloud patch on August 18th, 2019. (a) The object of a cloud patch at 13:57 UTC. (b)-(d) The effective radius (left) and the number of data samples (right) for the total, ice, and water cloud phases, respectively. (e)-(h) are the same as (a)-(d) but for the same cloud patch at 15:57 UTC.

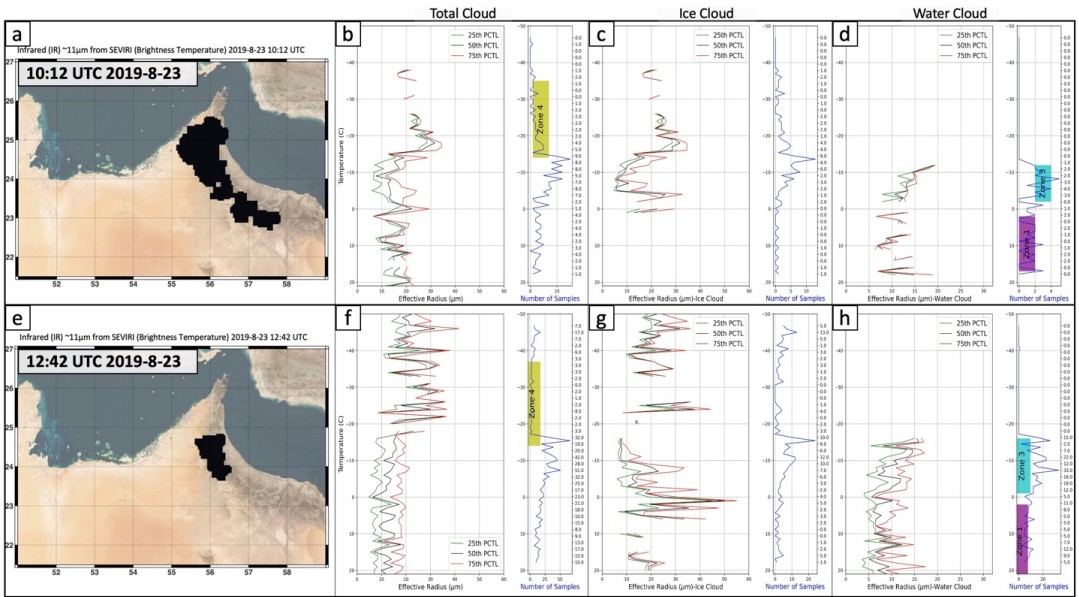

Figure 11. Same as Figure 10 but for a cloud patch at (a)-(d) 10:12 UTC and (e)-(h) 12:42 UTC on August 23rd, 2019, (SF07).



First, more examples of the zone detection at specific time points are provided. Figure 10 shows the identification of the zones at two different time points for a developing cloud on August 18th, 2019. In Figure 10a, the black object in the top panel is the cloud patch from

satellite data at 13:57 UTC, and Figures 10b-d show the identified Zones 1 and 5 from the satellite data of water and ice clouds, respectively. After 2-hour development, this cloud patch covers a larger area at 15:57 UTC (Figure 10e), and in addition to Zones 1 and 5, Zones 2, 3, and 4 are also detected based on the water and total cloud data (Figures 10f-h). In addition to SF03, Figure 11 shows the identification of the zones for the cloud case of SF07 on August 23rd, 2019.

At 10:12 UTC, the cloud is mainly over the southwest side of the Al Hajar Mountains, and Zones 1, 3, and 4 are detected in the water and total cloud data (Figures 11a-d). After 2.5 hours, the cloud object becomes significantly smaller, but Zones 1, 3, and 4 still exist at 12:42 UTC (Figures 11e-h).

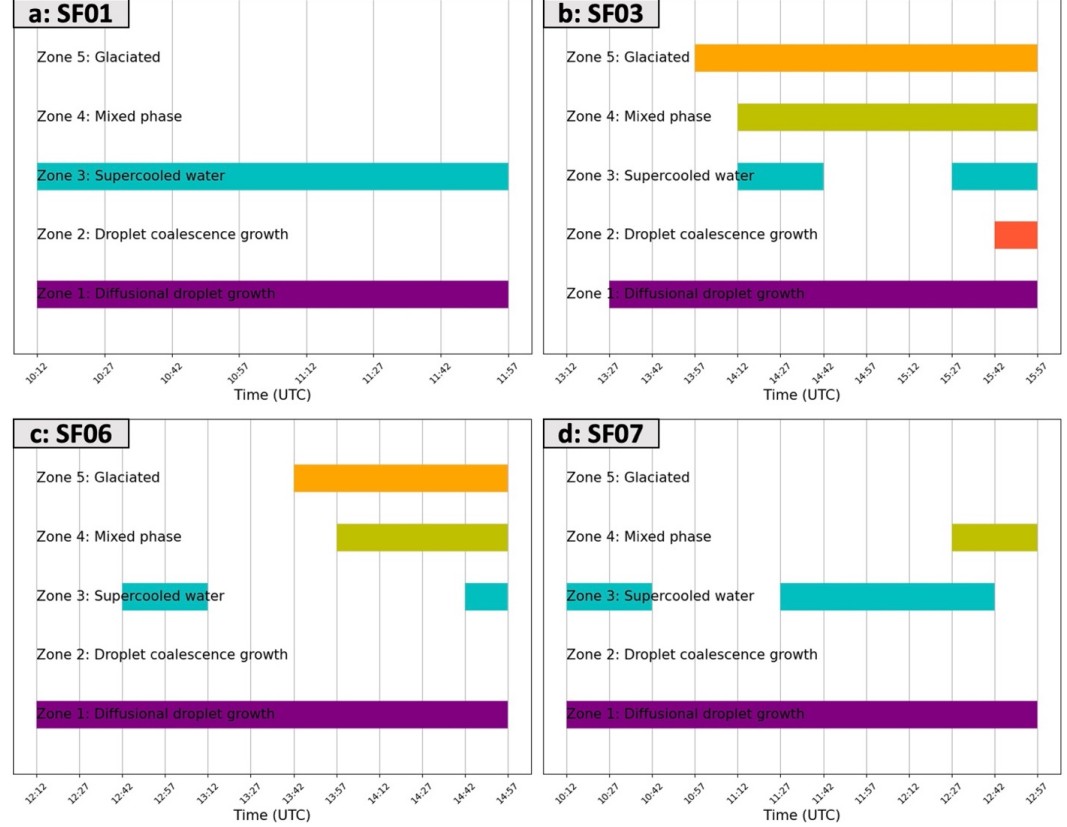






Figure 12. The evolution of the 5 zones in (a) cloud case SF01, (b) cloud case SF03, (c) cloud case SF06, and (d) cloud case SF07. The X-axis is time, and the Y-axis denotes the 5 zones.

The same detection processes can be repeated for any cloud patch through its life cycle so that we can examine the evolution of the cloud. To test the transferability of the 5-zone concept, the cloud zones are detected using the satellite data through the main time periods of the cloud cases SF01, SF03, SF06, and SF07 (Figure 12). These four cases were selected because they have sufficient satellite data to identify the zones and also aircraft measurements to validate the results (Figure 13). SF02 and SF04 are not presented here because of limited collocated aircraft

data samples and cloudy pixels/points in the satellite images to calculate the ER distributions for ice and water phases.

In the cloud case SF01 on August 12th, 2019, only Zones 1 and 3 are detected. These zones are detected continuously from 10:12 to 11:57 UTC (Figure 12a), indicating suppressed microphysical processes for this case. The cloud cases SF03 and SF06 have a similar evolution

of the zones, including continuous Zone 1, discontinuous Zone 3, and Zones 4 and 5 during the middle and later periods. The difference is that in the case SF03, Zones 4 and 5 develop earlier and exist longer, indicating that case SF03 has a more active ice production process. In case SF07, only Zones 1 and 3 are detected for a significant time period, suggesting suppressed microphysical processes for precipitation.

To validate the results of the zones based on satellite data, we examined the aircraft observation for those four cloud cases. Case SF03 is intensively examined in Section 3 as an example. Here, to conduct a comparison among the four cases, we selected an aircraft CP with a similar temperature (-12.1°C for the CPs from SF03 and SF07; -12.3°C for SF01; and -12.0°C for SF06) from each case and examined the cloud particle distribution and the 2DS images

(Figure 13). A relatively cold temperature (around -12°C) is selected because the main difference among those four cases based on our 5-zone framework is that SF03 and SF06 have Zones 4 and 5, but SF01 and SF07 do not, indicating a difference in the ice production process. The CPs around -12°C is close to the coldest observed temperature in all four flights, which presents the best validation for the difference of Zones 4 and 5. The cloud particle sizes from SF01 and SF07

are more concentrated at the relatively small size (Figures 13 a and d). They have a higher



concentration (close to or over $10 \times 10^3$ per liter per μm) than SF03 (below $3 \times 10^3$ per liter per μm) and SF06 (below $8 \times 10^3$ per liter per μm) from FCDP between 10-20 μm, which implies that the droplet growth in the cloud cases SF01 and SF07 is suppressed. The particle size distribution from 2DS and HVPS in SF03 has a long tail toward the large size, 100 – 1000 μm (Figure 13b),

while the distribution in SF06 indicates large particles around 100 μm (Figure 13c). The 2DS image examples in the right column of Figure 13 are in an agreement with the cloud particle distribution for particles greater than 20 μm. The 2DS image for SF03 shows many ice particles significantly larger than 100 μm, consistent with the detected continuous Zones 4 and 5 based on satellite data. The 2DS image of SF06 exhibits a few large particles around or larger than 100

μm, which has a fair agreement with the Zones 4 and 5 that exist for a relatively short period. The CPI images for the corresponding CPs from cloud cases SF01, SF03, and SF06 show similar results (right column in Figure 13). The CPI images capture the large ice particles in SF03 and SF06, while the image for SF01 does not have any large particles. The CPI image for SF07 is not included since it does not have any images available around the time of that CP (within 10

seconds before or after that CP).

      Overall, the 5-zone framework works for all four cloud cases, and the aircraft observation supports the identified zones based on satellite data.







Figure 13. (a) the distributions of cloud particle size (left) and 2DS images (top right) and CPI images (bottom right) for the cloud penetration at the temperature of -12.3°C in SF01. (b) same as (a) but for the cloud penetration at the temperature of -12.1°C in SF03. (c) same as (a) but for the cloud penetration at the temperature of -12.0°C in SF06. (d) same as (a) but for the cloud penetration at the temperature of -12.1°C in SF07; there is no CPI image during that cloud

penetration.

## 5. Summary and Discussion

In this study, we investigated the cloud microphysical features for some cloud cases in the UAE using aircraft observation, introduced a 5-zone framework to identify the cloud

microphysical zones using satellite data, and validated the zones detected from satellite data with aircraft measurements. Our study aims to provide scientific support to develop an applicable framework to examine cloud microphysical processes and detect suitable cloud features that could be targeted for precipitation enhancement in the UAE. A summary of this study is listed below.

1. The UAE 2019 Airborne Campaign provides a unique aircraft sensor dataset, which is analyzed to examine the microphysical features of some cloud cases in the UAE.

2. The effective radius (ER) retrieved from satellite data is in fair agreement with the ER measured by aircraft, adding confidence in using ER data from satellites to analyze the cloud microphysical features.

3. Following Rosenfeld and Lensky (1998), a new 5-zone framework was developed to identify the cloud microphysical zones using satellite data, which can be used to indicate the cloud microphysical processes and rainfall enhancement potential.

4. The 5-zone framework can successfully detect the cloud microphysical zones, including the glaciated zone with large ice particles. The results were validated with the aircraft

measurements for four cloud cases.

In addition to satellite data, radar data is often used to examine the impacts of cloud seeding (e.g., Vujovic and Protic, 2017; Zaremba et al., 2024; Wang et al., 2021). Meanwhile, the radar data might be a potential data source providing additional information to refine the detection of microphysical processes. We considered radar reflectivity as a supplementary data



source to characterize the cloud features related to precipitation. However, only the radar in Al
       Ain (Figure 1) overlaps with the observation area of three research flights (SF03, SF06, and
       SF07) and offers continuous vertical reflectivity profiles. We explored the potential relationship
       between the radar reflectivity and the cloud's microphysical features, as summarized in the
       Supplements. Due to the limited number of available samples, it is difficult to connect the radar
data and the cloud microphysical zones. More studies are needed to investigate the potential
       usage of radar data in detecting the cloud microphysical zones.

       The 5-zone framework in this study is similar to the 5-zone concept from Rosenfeld and
       Lensky (1998), but it uses a supercooled water zone instead of the rainout zone since this study
       focuses on the microphysical processes related to precipitation of convective clouds. In addition,
the thresholds of temperature, ER, ER growth rate, and the cloud phase are added for each zone
       in our framework. We used previous studies (Rosenfeld and Lensky, 1998; Wang et al., 2019) as
       conceptual references and the data from intensive analysis of real cloud case SF03 to determine
       those thresholds and then refined them through case study analysis of other cloud cases. Since all
       these cloud cases occur during summer in the UAE, the thresholds determined in this study are
considered specific for the summer (primarily convective systems) over the UAE. If this 5-zone
       framework is utilized for some different seasons or climate zones, a corresponding modification
       of those thresholds is needed due to the difference in the cloud microphysical features and the
       environmental aerosols. However, this framework presents a methodology that could be tuned to
       identify other cloud types using the threshold parameters identified.




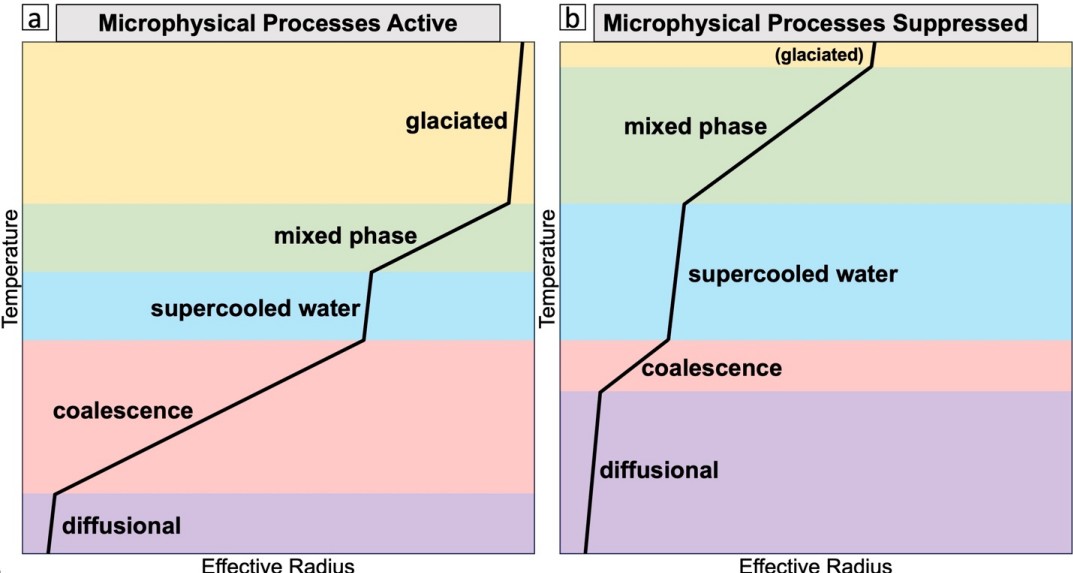

Figure 14. The schematic of the 5 zones for clouds with (a) active microphysical processes and (b) suppressed microphysical processes for precipitation.

The 5-zone framework presents a concept to identify the cloud microphysical zones and diagnose the cloud microphysical processes that affect precipitation. As the concept summarized in Figure 14, this framework focuses on the growth of cloud particle size, which can be quantified using effective radius retrieved from satellite data. While the microphysical processes favorable to precipitation initiation are active (Figure 14a), such as a deep coalescence zone or

the presence of a deep glaciated zone, precipitation may occur efficiently. On the other hand, the microphysical processes favorable to precipitation initiation are suppressed (Figure 14b), when a deep diffusional zone (the coalescence process is suppressed) is present or a deep supercooled water zone (the ice particle production process is not active) is present. In the 5-zone framework, we included thresholds to guide the mode of precipitation enhancement for the cases when

suppressed microphysical processes are detected, as shown in red at the bottom of Figure 8. The clouds with suppressed microphysical processes could be a glaciogenic (cold cloud) seeding target if Zone 3 is present and has sufficient depth, Zone 4 is present and has sufficient depth, or Zone 5 is absent, which indicates the ice particle production process is not active. On the other hand, it could be a hygroscopic (warm cloud) seeding target if Zone 1 is present and has

sufficient depth, Zone 2 is not present or is shallow, or Zone 3 is present, which indicates the



cloud droplet collision-coalescence process is suppressed. In addition to determining cloud seeding targets, this information about seeding types is advantageous in guiding cloud seeding operations.

In conclusion, this study has successfully introduced and applied a 5-zone framework to

identify cloud microphysical zones using satellite data, focusing on cloud microphysical processes related to precipitation and potential for rainfall enhancement in the UAE during summer. The performance of the framework was demonstrated through the analysis of cloud cases and validated with aircraft measurements. Future work will aim to enhance this approach by incorporating a machine learning-based cloud tracking algorithm applied to MSG data,

allowing for a more detailed examination of microphysical zones in near-real time within individually tracked cloud clusters. This advancement will further our understanding of cloud precipitation processes and improve the identification of suitable targets for precipitation enhancement.




...



**Data availability.** The ERA5 Reanalysis data can be found on the Climate Data Store website
from the Copernicus Climate Change Service, https://cds.climate.copernicus.eu/cdsapp#!/home.
The High Rate SEVIRI Level 1.5 Image Data and Optimal Cloud Analysis products are publicly
available through EUMETSAT Data Services, https://navigator.eumetsat.int/start.

**Competing interests.** The authors declare that they have no conflict of interest.


**Author contributions.** ZZ analyzed cloud microphysical features using aircraft observation
data, examined the meteorological conditions using ERA5 data, and wrote the manuscript with
contributions from all co-authors. VAG performed the analysis of cloud particle effective radius
using satellite data and developed the code to detect the five zones using satellite data. DA
supervised this study and guided the analysis. LDM supervised this study and reviewed the
manuscript. CR, EYK, and VC processed the radar data and helped with the interpterion of the
radar results.


**Acknowledgments.** This study is supported by the National Center of Meteorology (NCM), Abu
Dhabi, UAE, under the UAE Research Program for Rain Enhancement Science. The authors
acknowledge Dr. Paul Lawson and Dr. Brad Baker from the Stratton Park Engineering Company
(SPEC) for providing the aircraft observation data from the UAE 2019 Airborne Campaign. The
authors also acknowledge Dr. Youssef Wehbe and Dr. Michael Weston from NCM UAE for
their review and valuable comments on this manuscript.



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
