# Peer review of "An Analysis of Cloud Microphysical Features over UAE Using Multiple Data Sources"

_EGUsphere, 2024_

## Author Comment (AC1)

**Response to Reviewer 1**

This study proposes a methodology to infer precipitation-forming processes based on satellite retrievals of the vertical profiles of convective clouds' particle effective radius and phase. The satellite is the METEOSAT Second Generation (MSG) over the Indian Ocean. The target clouds are over the UAE. The objective is to determine the clouds' seedability based on conceptual models. The general idea and objectives are of great interest, but major methodological issues require a major revision before the paper can get accepted.

We appreciate the valuable comments from the reviewer.
Our responses are in blue below each comment from the reviewer.

**Major comments:**

The satellite retrieved cloud drop effective radius product (RE) has three types: water, ice, and total cloud. However, these retrievals, as presented here, raise many questions:

How can ice retrievals of RE exist for clouds with temperature > 0C (e.g., Fig. 7b)?

Why are the total and water cloud RE values different at T>0C, where ice cannot exist?

How can water retrievals of RE exist for clouds with temperature < -40C (e.g., Fig. 10d), where water cannot exist?

Why are the total and ice clouds RE different at T <-40C, where water cannot exist?

We thank the reviewer for their thorough review and for raising these methodological points. We appreciate the detailed comments and would like to address these points as follows:

First, we acknowledge the confusion regarding the presence of ice retrievals in conditions where temperatures > 0°C. To avoid confusion, we will remove all data points related to ice retrievals at temperatures above 0°C. This adjustment will ensure that our analysis remains within the physically realistic bounds. We also understand the concern regarding the differences between total and water cloud RE values at temperatures greater than 0°C, where ice should not exist. The ATBD (2016) explains that the retrieval algorithm may mistakenly attribute large CRE values to water clouds due to differential absorption characteristics in the 1.6 μm channel. As quoted from the ATBD: "The ice-water absorption differential in the 1.6 μm channel (daytime only, see section 3.1 of ATBD) means that a scene that contains ice clouds but is interpreted as water clouds would appear to have very large CRE values. These are identified as being larger than the physical limits set on the water CREs, and in this case, a switch to the ice phase is made. The reverse process can occur in the case that ice is assumed and in reality, water clouds are present". To mitigate this issue, we will include a filtering step to eliminate such anomalies in our revised analysis and explain this limitation in the manuscript.

Additionally, the differences observed between total and ice cloud RE at temperatures below -40°C suggest retrieval errors. As these cases are rare (only 1-3 samples of "Noise"), we will remove these outliers from our analysis to ensure that only physically consistent data is presented. We will also revise Figures 10c and 10g to reflect these corrections, ensuring that data plotted for ice clouds are limited to temperatures below 0°C and that any inconsistencies between total and phase-specific RE are addressed.

We believe these revisions will enhance the clarity and accuracy of our study, and we appreciate your constructive feedback.

The methodology depends critically on the definitions of the microphysical zones, as defined in Figure 8. The cloud phase is used for the zones' definitions, but it is not explicitly given by the OCA EUMETSAT cloud data, as all three types (water, total, and ice) have the effective radius (RE) at all temperatures.

Zone 1, diffusional droplet growth is defined by water phase cloud at T>0C and RE < 10 um. However, clouds can have diffusional growth at temperatures well below 0C, and at RE >10 um, until significant coalescence starts increasing RE beyond the drop growth rate by condensation only. This happens at RE>12 um, or even higher. A case in point is clouds with cold bases, as typical to the UAE. The lower parts of these clouds must be in the diffusional growth zone, as they are composed of very small water droplets.

We agree with the reviewer's comment about diffusional growth at ER > 10 μm and at temperatures < 0°C. We selected the ER and temperature thresholds for the diffusional zone based on the cloud penetration ER in the UAE during the summer (2019 August). Figure 3 of the manuscript shows that the mean effective radius is always < 10 μm at temperatures > 0°C. Our choice of diffusional droplet growth at T > 0°C and RE < 10 μm is to identify extreme cases of diffusional growth and to also eliminate cases when coalescence may be marginally active. This is to identify the best targets for cloud seeding using hygroscopic particles, which is most effective in cases when coalescence is not active.

Zone 2, droplet coalescence growth zone. The description in Figure 8 has details that are not described in the text (lines 494-500) and are not understandable. What is T21? Why is the 75th percentile taken and not the 50th percentile? Anyway, the 25th percentile should be the relevant number because this pertains to the smaller ER in the growing convective elements. In Figure 8 there is the condition of 20>ER>15 um. There seems to be an uncovered gap between Zones 1 and 2.

The reviewer accurately points out that we did not describe all the details in Figure 8 and we intend to correct that in the revised manuscript. $T_{Z1}$ is the coldest temperature of zone 1, which means the warmest temperature of zone 2 should be lower than the coldest temperature in zone 1 but higher than -10°C. This is done to create a separation between zone 1 and zone 2, so that clouds with active coalescence are not targeted for hygroscopic seeding. We will add the relevant description text in the revised manuscript.

We use the 75[th] percentile because we want to identify the relatively large droplets developed by collision and coalescence. Although there are still many small droplets, some droplets may start to grow quickly due to coalescence, which will result in a long tail in the droplet size distribution. The figure below shows an example from one cloud penetration at temperature -5.2°C from SF03. The 75[th] percentile is more sensitive to those large droplets compared to the 50[th] percentile.

Based on the ER threshold, there is a gap between zones 1 (50[th] ER < 10μm) and 2 (15μm < 75[th] ER < 20μm), but the ER is only one of the metrics to identify these zones. In addition, the selection of those thresholds for different zones in this study aims to provide guidance for

detecting cloud seeding targets, determining if a cloud patch is a suitable seeding target or not. Although there might be some small gap between different zones, it will not impact the detection of those microphysical zones that are intended for the cloud seeding application.

[Figure]

*Figure: The distribution of cloud particle effective radius for one cloud penetration at temperature -5.2°C from flight SF03. The three red vertical bars from left to right indicate the 25th, 50th, and 75th percentiles of the effective radius.*

Zone 3, Supercooled water zone. Why should the supercooled water zone be independent of the diffusional growth or coalescence zones? Both can occur at temperatures < 0 and contain supercooled water. It just does not make sense in both logical and physical ways. There is a potential overlap in the conditions for the zones because the same scene can have conditions that fulfill both Zone 2 and 3. This should not happen. The extension of mixed-phase to temperatures colder than -35 C in all presented cases with identification of that zone is unreasonable physically. Such cold glaciation temperatures were documented by aircraft only in storms with severe updrafts that caused large hail. Furthermore, VIIRS views of similar clouds in the same region generally resulted in a glaciation temperature of -20 C, due to the dusty conditions. This determination is based on the unique channel combination of VIIRS that allows us to detect unambiguously the glaciated clouds.

The algorithm development is motivated by the requirements of the cloud seeding operator to conduct hygroscopic and glaciogenic seeding. In glaciogenic seeding, the operators will target supercooled water, even when this supercooled region overlaps with the presence of high coalescence activity.

In addition, our algorithm separates zone 3 and zone 2 using droplet growth rates (dER/dT). Since supercooled drops are more likely to develop in the absence of coalescence, the growth rate in the supercooled zone is defined as being smaller than that in the coalescence zone as shown in Figure 8.

In the revised manuscript we will analyze the glaciation temperature in VIIRS channels as suggested by the reviewer. This analysis will give us a more accurate representation of glaciation

temperature in zones 3, 4, and 5, given that the aircraft data did not sample clouds cold enough to reach this level.

Zone 4, Mixed phase zone. The mixed phase is defined only for clouds colder than -10C, but it can occur at higher temperatures. According to the text, the total cloud ER is used here, but this is not mentioned in Figure 8. Again, since different kinds of ER are used for zones 2 and 3, there can be much overlap in conditions not recognized here, causing definition ambiguity.

We agree that the mixed phase could occur at a temperature higher than -10°C. However, based on the aircraft observation in this UAE campaign in August of 2019, the warmest temperature when ice particles appear is at least -10°C. Therefore, we use -10°C as the threshold.

We will add the "total cloud" product in the box of zone 4 in Figure 8.

Yes, zones 2 and 3 may have some overlap if only based on ER thresholds. However, as we mentioned above, we used different droplet growth rate thresholds to differentiate between zone 2 and zone 3.

Zone 5, Glaciated cloud. Its definition relies on the ice ER. But ice ER seems to be unreliable in many ways. It exists for warm clouds with T>0C. In Figure 10g it has the same large ice ER>25 um at all temperatures. Its ER in some cases (e.g., Fig 10c) is extremely small, much smaller than is almost ever detectable at -45 C by MODIS.

We understand the concern that the cloud phase, critical for defining the microphysical zones, is represented by all three types (water, total, and ice) with an effective radius (RE) at all temperatures. This could lead to confusion, particularly in the Zone 5 (Glaciated cloud) definition. As explained by the OCA ATBD (please see the figure below that shows the distribution of effective radius for the ice phase clouds), most of the population contributes to the 23 μm peak, while a peak is observed at 5.0 μm for the water phase. However, some noise in the data is explained in the algorithm documentation, highlighting limitations related to broken or sub-pixel clouds, highly heterogeneous clouds, and mixed-phase clouds.

To address this, we will revise the definition of Zone 5 (Glaciated cloud) and filter out data for temperatures greater than 0°C, as discussed in our previous response, to ensure it aligns with the expected physical characteristics. Specifically, we will reassess the reliance on the ice ER and consider alternative criteria for defining this zone, particularly given the potential unreliability of the ice ER at specific temperatures.

Additionally, in the manuscript, we will provide a detailed explanation of the anomalies observed, such as the large ice ER values across all temperatures and the minimal ice ER values at very low temperatures. This explanation will draw on insights from the OCA product guide and relevant literature to contextualize these findings and ensure a clear understanding of the data presented.

[Figure]

*Figure: Histograms of CRE associated with ice phase clouds.*
*This figure is from the EUMETSAT Optimal Cloud Analysis: Product Guide (online):*
*https://user.eumetsat.int/s3/eup-strapi-media/Optimal_Cloud_Analysis_Product_Guide_366f360a7c.pdf*

The criteria selection is based on and validated by comparing aircraft in situ cloud measurements. However, the aircraft did not penetrate clouds colder than -13C, so there is no aircraft validation to the zones below that temperature. But the authors claim, and rightly so, that seeding potential is best at clouds with supercooled water that extend deepest, which pertains mostly to the unvalidated temperature range.

Therefore, I recommend that the authors calibrate and validate the MSG retrievals against VIIRS and then redo all their calculations and revise their inferences as necessary.

We appreciate the reviewer's valuable suggestion. We will proceed with this analysis if the data coincides in time and space with aircraft data. If the data does not coincide, we can still perform an analysis of glaciation temperature to adjust thresholds in zones 3, 4, and 5.

**Minor comments:**

Line 87: Growth of precipitation particles cannot occur "through collision and coalescence of the ice multiplication process".

We will fix the error (changing "of" to "or"), and the new sentence in the revised manuscript is "Growth of precipitation particles can either occur through collision and coalescence or the ice multiplication process or a combination of the two."

Line 88: Raindrops cannot form by diffusional growth alone in convective clouds with any cloud base temperature.

We will modify this sentence as suggested: "Raindrops cannot form by diffusional growth alone in convective clouds."

Line 371: Please be aware and discuss the gap between the satellite-retrieved rain threshold of 15 um (e.g., Lensky and Shiff, 2007) and the aircraft-retrieved threshold of 12 um (Freud and Rosenfeld, 2012).

We will add a couple of sentences to discuss the gap between the satellite-retrieved rain threshold (e.g., Lensky and Shiff, 2007) and the aircraft-retrieved threshold (Freud and Rosenfeld, 2012) as suggested.

---

## Author Comment (AC2)

**Response to Reviewer 2**

This paper introduces a framework that categorizes convective cloud features from UAE into 5 different microphysical zones using satellite data. The effectiveness of this framework is then evaluated using aircraft observations from UAE 2019 Airborne Campaign. Overall, this paper is easy to follow. However, while the framework is somewhat interesting, there are several flaws in the methodology that requires some re-work.

We appreciate the valuable comments from the reviewer.
Our responses are in blue below each comment from the reviewer.

**Major comments:**

1. If I am understanding correctly, this paper targets the microphysics of continental convective clouds. However, it is surprising that for deriving the in situ measured cloud droplet size distribution and cloud effective radius (Reff), this paper is only using the FCDP. The author claims that this choice is based on the fact that the LWC derived from FCDP has higher correlation with the Nevzorov probe measured LWC. However, if you are studying the different microphysical processes (diffusional growth vs. collision coalescence), are having both liquid and ice phased clouds in your sampling, it is almost common practice to combine the FCDP/FFSSP with 2DS (or even HVPS), (see Rosenfeld and Lensky 1998, Painemal and Zuidema 2011, Kang et al. 2021). By using FCDP only, the in situ Reff will be biased towards the smaller droplets.

Rosenfeld, D., and I. M. Lensky, 1998: Satellite-Based Insights into Precipitation Formation Processes in Continental and Maritime Convective Clouds. Bull. Amer. Meteor. Soc., 79, 2457–2476, https://doi.org/10.1175/1520-0477(1998)079<2457:SBIIPF>2.0.CO;2.

Painemal, D., and P. Zuidema (2011), Assessment of MODIS cloud effective radius and optical thickness retrievals over the Southeast Pacific with VOCALS-REx in situ measurements, J. Geophys. Res., 116, D24206, doi:10.1029/2011JD016155.

Kang, L., Marchand, R. T., & Smith, W. L. (2021). Evaluation of MODIS and Himawari-8 low clouds retrievals over the Southern Ocean with in situ measurements from the SOCRATES campaign. Earth and Space Science, 8, e2020EA001397. https://doi.org/10.1029/2020EA001397

We appreciate the reviewer's suggestion and the reference. We agree with the reviewer and will recalculate the effective radius (ER) in this study using the combined cloud particle size distribution from FCDP, 2DS, and HVPS. Following Fu et al. (2022), we use 40 μm as a fixed break point to combine the FCDP and 2DS particle size distribution. The break point between 2DS and HVPS is 1000 μm. Only size distributions with total number concentrations greater than 10 cm$^{-3}$ are included in calculating ERs (Fu et al., 2022). According to our tests, the new ERs based on combined size distribution (FCDP, 2DS, and HVPS) tend to be slightly larger (around or below 5%) than the ERs based on FCDP for most cloud penetrations (CPs). Because the large particle size (>40 μm) concentration from 2DS is much lower compared to the particle size concentration from FCDP, and the concentration from HVPS is extremely low (close or equal to 0) for most CPs (e.g., Figures 6 and 13 in the manuscript). Meanwhile, the difference between the new ERs and FCDP ERs becomes larger for those CPs at a cold temperature (e.g., around -12°C), which might be related to the appearance and increase of ice particles. In the

revised manuscript, we will recalculate the ER, add a paragraph to describe the calculation of ER in Section 2 Dataset and Methodology, and modify the text accordingly throughout the manuscript.

*Fu, D., Di Girolamo, L., Rauber, R. M., McFarquhar, G. M., Nesbitt, S. W., Loveridge, J., ... & Scarino, A. J. (2022). An evaluation of the liquid cloud droplet effective radius derived from MODIS, airborne remote sensing, and in situ measurements from CAMP 2 Ex. Atmospheric Chemistry and Physics, 22(12), 8259-8285.*

2. The authors use the in-situ probe measured Reff to characterize the cloud microphysics, but then use the satellite measured Reff to build the framework to identify the different microphysical zones of convective clouds. This works under the assumption that the satellite measurements are representative of the in-situ measurements. However, the authors need to keep in mind that passive satellite retrieved Reff is a cloud top effective radius (which is not vertically resolved in any way), and the in situ Reff could be sampled from anywhere within a cloud (depending on where the cloud pass/penetration took place). These are conceptually different, and the authors need to be careful when using the different Reffs to characterize cloud microphysical processes.

We thank the reviewer for the valuable feedback. We acknowledge that these two measurements represent slightly different physical quantities, with the in-situ measurements capturing the microphysical properties at specific levels within the cloud, while the satellite retrievals provide a broader, less vertically resolved perspective. In our study, we used the satellite-derived ER to build the framework for identifying the microphysical zones of convective clouds, assuming that the satellite measurements can provide a consistent, albeit generalized, representation of the cloud's microphysical state. However, we agree that this assumption requires careful consideration, especially when compared with in situ data that offers a more detailed view of the cloud's vertical structure.

To address this, we will revise the manuscript to more explicitly discuss the limitations and assumptions of using satellite-derived information to characterize cloud microphysics. We will highlight the differences between the cloud-top measurements from satellites and the within-cloud measurements from in situ probes, emphasizing the implications for identifying microphysical zones. Additionally, we will include a more detailed discussion of the potential discrepancies and how they might affect the interpretation of the cloud microphysical processes.

3. I don't think the authors elaborated on how the collocation between the aircraft measurements and the satellite observations were done.

We appreciate the opportunity to elaborate on this aspect of our methodology.

The collocation between the aircraft measurements and satellite observations was carefully conducted to ensure the accuracy and relevance of our comparative analysis. The process will be described in more detail in Section 3 of the manuscript. We will detail the steps taken to match the aircraft cloud penetration data with the corresponding satellite data. To summarize:

- Temporal Collocation: We applied a temporal threshold of 5 minutes between the aircraft cloud penetrations and the satellite observations to ensure that the measurements were as

close in time as possible. This was necessary because of the 15-minute temporal resolution of the products from the SEVIRI sensor onboard the MSG satellite.

- Spatial Collocation: For spatial collocation, we applied a spatial threshold of 3 km, which allowed us to match the cloud features observed by the aircraft with the satellite data. The aircraft data provides a high-resolution, detailed view of specific cloud penetrations, and this proximity ensures that the satellite data used for comparison represents the same cloud features.

- Verification of Collocation: The collocated data points were further verified by comparing the retrieved ERs from both the satellite and aircraft data. As shown in Figure 7 of the manuscript, the satellite-derived ER values were compared with those measured by the aircraft, and we found that the aircraft-measured ER values generally corresponded well with the satellite data, particularly when considering the differences in measurement techniques and resolutions.

4. The thresholds used in the framework is somewhat confusing. What is the rational of choosing the 50th percentile Reff or the 25th percentile Reff in the different zones? The mixed-phase zone also seems to overlap quite a bit with the collision-coalescence zone and the supercooled water zone. Is it really necessary to have 5 different microphysical zones?

We used the 50th percentile of ER for zones 1 and 3, but used the 75th percentile of ER for zones 2, 4, and 5 to focus more on the relatively large cloud particles. For zones 2 and 4, although there are still many small droplets, some particles may start to grow quickly due to coalescence (droplet growth in zone 2) or ice particle formation (ice growth in zone 4), which will result in a long tail in the particle size distribution. The figure below shows an example from one cloud penetration at temperature -5.2°C from SF03. The 75th percentile is more sensitive to relatively larger droplets compared to the 50th percentile and indicates that collisions are producing larger droplets. Therefore, we use the 75th percentile of ER for those zones when we need to detect a tail in the particle size distribution.

The mixed-phase zone is colder (-10°C > T > -38°C) and has larger particles (75th ER > 20µm) compared to the coalescence zone, which is warmer than -10°C and has relatively smaller particles (75th ER < 20µm). The difference between the mixed-phase zone and the supercooled water zone is the particle growth rate. The growth rate of the mixed-phase zone is larger than the growth rate of the supercooled water zone as shown in Figure 8. Overall, the 5 zones have different physical features, and their definitions are motivated by the requirements of the cloud seeding operator to conduct hygroscopic and glaciogenic seeding as labeled in Figure 8. Therefore, we defined 5 different microphysical zones.

[Figure]

*Figure: The distribution of cloud particle effective radius for one cloud penetration at temperature -5.2°C from flight SF03. The three red vertical bars from left to right indicate the 25th, 50th, and 75th percentiles of the effective radius.*

**Minor comments:**

Line 28: precipitation-producing clouds -> precipitating clouds

We will revise it as suggested.

Line 208: Any possible reason why the LWC from FFSSP has a much lower correlation with the LWC from Nevzorov?

Based on our comparison of the data obtained in this UAE airborne campaign, the LWC from FFSSP has a lower correlation with the LWC from Nevzorov compared to the LWC from the FCDP. Possible reasons for the difference in correlation could be related to sampling bias and/or instrument response, as well as the collection efficiency of droplets in the Nevzorov probe. In this study, we focus on the comparison of the FCDP and FFSSP, both forward scattering spectrometers, with a different instrument that uses other physical properties of droplet sizing, in this case the hotwire mechanism in the Nevzorov LWC. The finding that the FCDP correlates better with the Nevzorov gives us confidence that the FCDP provides higher accuracy compared to the FFSSP.

Line 210: It is surprising that you are using FCDP alone to derive effective radius for convective clouds…

As we responded to the major comment 1, we will recalculate all the effective radii in this manuscript using the combined particle size distribution from FCDP, 2DS, and HVPS. Please see our response to major comment 1 for more details.

Line 220: How is this "mean effective radius" defined? Is the range indicating the range of "mean effective radius for each CP"?

In each cloud penetration, there are one to several observed values (one observed value per second). The mean effective radius here is defined as the average effective radius of all observed values in each cloud penetration. We will add one sentence to clarify in the revised manuscript.

Line 371: If I am understanding this correctly, this 15 μm threshold was derived from AVHRR measurements, is this applicable to the in situ measured Reff?

The 15 μm precipitation threshold in convective clouds from Lensky and Drori (2007) and Lensky and Shiff (2007) was derived from satellite measurements. Meanwhile, Freud and Rosenfeld (2012) defined 14 μm as the precipitation threshold for warm rain in convective clouds based on aircraft-retrieved data. We will revise those sentences to include these citations in the revised manuscript.

Line 377: (d) total water content from Nevzorov.

We will revise it as suggested.

Line 450: why are there ice cloud effective radius retrievals at T~ 10°C from in situ probes?

The effective radii from aircraft observation (black dots) in Figure 7 are all the effective radius values obtained from the in-situ probes during flight SF03, and they could be from water or ice clouds. The total, water, and ice cloud labels in panels a-c are specifically for the satellite data. We will modify the caption of Figure 7 to clarify.

Line 464: why choosing to use FCDP to derive cloud effective radius when you are targeted at convective clouds?

As we responded to the major comment 1, we will recalculate all the effective radii in this manuscript using the combined particle size distribution from FCDP, 2DS, and HVPS. Please see our response to major comment 1 for more details.

Line 468 to 470: I cannot agree with the statement that "the ERs from aircraft and satellite datasets have a fair agreement", your Figure 7 is suggesting otherwise…which really questions the validity of using ERs from satellite data to build the different cloud zones.

The aircraft ERs in Figure 7 are calculated from FCDP particle size distribution, which measures the size of particles in the 2-50 μm diameter range and is sensitive to water droplets. Therefore, they tend to be close to the water cloud ERs from satellites but smaller than the ice cloud ERs due to the lack of sensitivity of the FCDP to ice particles. As we mentioned in our response to major comment 1, we will recalculate all the ERs in this manuscript using the combined particle size distribution from FCDP, 2DS, and HVPS. Based on some tests, the difference between the new ERs and the ice cloud ERs from the satellite is a bit smaller, although there are still some differences between the ERs from aircraft and satellite. Some previous studies (e.g., Rosenfeld

and Lensky, 1998) found similar differences between the ERs measured by aircraft in-situ probes and satellites. In the figure below, the aircraft FSSP ERs (empty circles) tend to be smaller than the satellite ERs, which is similar to our result in Figure 7.

*Rosenfeld, D., & Lensky, I. M. (1998). Satellite-based insights into precipitation formation processes in continental and maritime convective clouds. Bulletin of the American Meteorological Society, 79(11), 2457-2476.*

[Figure]

*Figure 11b from Rosenfeld and Lensky, 1998. The 10th, 25th, 50th, 75th, and 90th percentiles of the effective radius. The median is indicated by the thick line. The vertical bars denote the different microphysical zones as numbered in the text. The aircraft FSSP effective radius measurements are denoted as empty circles, and the CLWC (g m^-3) are in plotted black circles.*

Line 497: Why in some of the zones the ER thresholds are using the 75th percentile, and in others the 50th percentile is used?

The criteria used in the 5 zone framework is somewhat ambiguous. If I am understanding this correctly, the authors are trying to formulate the 5 zone framework using thresholds of ER, dER/dt, BT, wouldn't this cause some overlap between different zones? I would suggest sticking with the same percentile (whether 25th, 50th, or 75th) and be consistent.

We use the 75th percentile for zones 2, 4, and 5 because we want to detect the relatively large particles in the tail of the particle size distribution that form by coalescence or ice formation. The figure in the response to major comment 4 shows an example from one cloud penetration at temperature -5.2°C from SF03. The 75th percentile is more sensitive to those large particles compared to the 50th percentile.

There might be some gaps between different zones. However, the detection of these zones in this study aims to provide guidance for detecting cloud seeding targets, determining if a cloud patch is a suitable seeding target or not. Although there might be some small gap between different zones, it will not impact the detection of those microphysical zones that are intended for the cloud seeding application.

Line 540: Figure 9 is suggesting that water cloud Reff is used in zone 1 and zone 2 and zone 3, total cloud Reff is used in zone 4, ice cloud Reff is used in zone 5. This was not mentioned in the main text.

We mentioned using different cloud phases to detect different zones in the definition and identification of each zone at lines 488-515 and in the description of Figure 9 at lines 530-535.

---

## Author Comment (AC3)

**Response to Reviewer 3**

This study combines in situ aircraft data collected in summertime convective clouds over the UAE in 2019 with satellite data, ground radar, and reanalysis to adapt the development of a 5 microphysical zone categorization to said region and season. The 5 zones are: (1) diffusional droplet growth zone, (2) droplet coalescence growth zone, (3) rainout zone, (4) mixed-phase zone, and (5) glaciated zone. A case study is then presented for evaluation. The study presents a novel adaptation of the proposed categorization that presents a useful framework for future analyses. Said framework is specific to the season and environment of the aircraft data, however, so further efforts should be made to better categorize that environment to improve potential applicability to other seasons and regions. With revisions, the manuscript could be suitable for publication.

We appreciate the valuable comments from the reviewer.
Our responses are in blue below each comment from the reviewer.

**Specific Comments:**

1. The cloud droplet effective radius looks to be determined solely from the FCDP, when it should be determined as a composite from multiple cloud probes, not just the FCDP. Larger particles seen by the 2DS (and HVPS) should also be included, for both the liquid and the ice clouds. Discussion following from line 465 suggests that the FCDP is not sensitive to ice particles, when in fact it does measure ice crystals, albeit with more uncertainty in sizing than of water droplets. However, were the 2DS and HVPS measurements in the ice clouds included in the calculation of effective radius, one might anticipate improved agreement between the in situ and satellite estimates. The current statement that they disagree because the FCDP is insensitive to ice particles is insufficient.

We agree with the reviewer that there is value in including the 2DS and HVPS in the calculation of effective radius. As such, we plan to recalculate the effective radius (ER) in this study using the combined cloud particle size distribution from FCDP, 2DS, and HVPS. Following Fu et al. (2022), we use 40 μm as a fixed break point to combine the FCDP and 2DS particle size distribution. The break point between 2DS and HVPS is 1000 μm, given that the 2DS is a 10 μm resolution optical array probe and the HVPS is a 100 μm resolution optical array probe. Only size distributions with total number concentrations greater than 10 cm$^{-3}$ are included in calculating ERs, to make sure that the cloud is not impacted by entrainment as cloud edges. According to our tests, the new ERs based on combined size distribution (FCDP, 2DS, and HVPS) tend to be slightly larger (around or below 5%) than the ERs based on FCDP for most cloud penetrations (CPs). Because the large particle size (>40 μm) concentration from 2DS is much lower compared to the particle size concentration from FCDP, and the concentration from HVPS is extremely low (close or equal to 0) for most CPs (e.g., Figures 6 and 13 in the manuscript). Meanwhile, the difference between the new ERs and FCDP ERs becomes larger for those CPs at a cold temperature (e.g., around -12°C), which is related to the increase of ice particles. In the revised manuscript, we will recalculate the ER to include the 2DS and HVPS, add a paragraph to describe the calculation of ER in Section 2 (Dataset and Methodology), and modify the text accordingly throughout the manuscript.

*Fu, D., Di Girolamo, L., Rauber, R. M., McFarquhar, G. M., Nesbitt, S. W., Loveridge, J., ... & Scarino, A. J. (2022). An evaluation of the liquid cloud droplet effective radius derived from MODIS, airborne remote sensing, and in situ measurements from CAMP 2 Ex. Atmospheric Chemistry and Physics, 22(12), 8259-8285.*

2.  The analysis of the in-situ data is lacking in qualifying the environmental conditions and the cloud evolutionary stage of the selected cloud passes. For example:

Lines 591-594: Do we know that they are not all young turrets being sampled in SF07?  Is there any indication of cloud age or development stage of these case studies?

The aircraft typically targets growing young turrets in the early stage of the cloud lifetime because they are most valuable to sample for studying the formation of precipitation and they are also safer to penetrate. Young growing turrets are also most valuable as targets for cloud seeding, which is the main application for developing the 5 microphysical zone characterization in this study. We reviewed the video of those flights and confirmed that the clouds sampled are young targets in the early- to mid-life cycle convection. The figure below is an example from flight SF03 showing that the aircraft was penetrating a relatively young turret.

[Figure]

*Figure: Image from the aircraft video in flight SF03 on August 18, 2019. The aircraft was penetrating a relatively young cloud turret at 13:41:51 UTC. The temperature measured by the aircraft was -5.7°C at that time.*

In addition, we also examined the vertical profiles of radar reflectivity associated with those CPs (at the same time and same location as those CPs), which suggests that these clouds are relatively young in their lifetime. The figure below shows the vertical profiles of radar reflectivity for the CPs in SF01 and SF03. The radar reflectivity ranges from 0 to ~30 dBZ, indicating characteristics of early- to mid-life cycle convection. Our radar observations do not show the high reflectivity (higher than 40 dBZ) typical of mature convection (Zipser et al., 2006; Feng et al., 2018).

*Zipser, E. J., D. J. Cecil, C. Liu, S. W. Nesbitt, and D. P. Yorty, 2006: Where Are The Most Intense Thunderstorms on Earth? Bulletin of the American Meteorological Society, 87, 1057–1072, https://doi.org/10.1175/BAMS-87-8-1057.*

*Feng, Z., Leung, L. R., Houze, R. A., Jr.,Hagos, S., Hardin, J., Yang, Q., et al., 2018: Structure and evolution of mesoscale convective systems: Sensitivity to cloud microphysics in convection-permitting simulations over the United States. Journal of Advances in Modeling Earth Systems, 10, 1470–1494. https://doi.org/10.1029/2018MS001305.*

[Figure]

*Figure: The vertical profiles of radar reflectivity associated with the CPs in SF01 (left) and SF03 (right) from the C-band weather radar at Al Ain. Each trace corresponds to the vertical profiles of radar reflectivity (above and below the aircraft) at the same time and same location (Lat & Lon) for each CP.*

Lines 597-599: Do we know if the cloud passes at these levels were in a similar age of cloud life cycle? Were they at similar distance from cloud top and cloud base?

As we mentioned above, the aircraft typically targets convection in the early- to mid-life cycle. The aircraft usually penetrates the cloud near the cloud top, usually 1000 ft below the top of growing turrets. Since this procedure was followed consistently during the experiment, it can be assumed that their distance from cloud tops and bases across the cases are fairly similar.

Line 608: "...implies that the droplet growth in the cloud cases SF01 and SF07 is suppressed", are the cloud passes under consideration consistent enough to make this conclusion? What were the cloud base temperatures? What are the cloud top temperatures/heights? What are the environmental conditions for the various days?

Based on the CPs observed by aircraft, those cloud cases have similar cloud top temperatures as shown in the table below; for the cloud bottom, SF06 and SF07 have a relatively colder cloud base (lowest CP) compared to SF01 and SF03. As discussed above, the life cycles of those convective cloud cases are similar, at the early- to mid-life cycle.

|  | Top Alt (ft) | Top T (C) | Bottom Alt (ft) | Bottom T (C) |
|---|---|---|---|---|
| SF01 | 23121 | -13.5 | 12018 | 9.3 |
| SF03 | 22750 | -13.0 | 11674 | 9.2 |
| SF06 | 23294 | -16.1 | 15983 | -3.3 |
| SF07 | 20155 | -12.1 | 13403 | 1.8 |

*Table: the cloud top (highest CP) and cloud base (lowest CP) for the four cases (SF01, SF03, SF06, and SF07) according to the definition of a CP in the aircraft data.*

Figure 12 and associated case studies: were the temporal measurements of these cloud cases all from the same convective turret, or could they have been different turrets (differing potentially in cloud top height, cloud age, etc.)?

During the flight, if the turret was growing, then the turret was profiled vertically through subsequent cloud penetrations in vertical increments in its early- to mid-life cycle stage. Once the cloud top was reached and the cloud transitioned to the past mid-life cycle (or rainout stage), a different and younger cloud target within the same cloud field was selected. This can be distinguished from a change in the altitude of the aircraft within a relatively small radius from the previous location. We also checked the video from the flights to confirm that.

Line 603 states the coldest observed temperature was -12 C in all four flights. Is this the coldest temperature because it was near cloud top, or was there another sampling reason? How close was sampling performed to cloud top?

The CPs at -12°C in those flights are close to the coldest CP observed by the aircraft. Satellite imagery shows colder temperatures, but those colder clouds were not penetrated because they

usually transitioned to a mature life cycle stage. As mentioned above, the aircraft typically targets the clouds at the early- to mid-life cycle stage.

All of the utilized cloud passes should be better qualified to improve usefulness and applicability of the analysis, and perhaps the analysis revised to include only cloud passes of comparable nature (it is currently unclear if they are comparable or not from the lack of context for the chosen cloud passes).

These clouds are comparable because the aircraft videos confirm that they are within the same cloud field and sampled by the aircraft in their early- to mid-life cycle stage. In addition, the radar reflectivity of those CPs confirms that those CPs are relatively young. We will include a few snapshots from the aircraft videos and a description of the cloud lifetime to clarify the cloud environment in the revised manuscript.

**Minor Comments:**

Figure 10. and 11. are cramped and very difficult to read

We agree with the observation regarding Figures 10 and 11. To improve readability, we will revise these figures by increasing the spacing and font size and reorganizing the layout to ensure all elements are clear and easy to interpret. We will include the updated figures in our revised manuscript.

Figure 12. The font of the x-axis needs to be bigger to be able to read the times.

We will revise the figure to increase the font size of the x-axis in our revised manuscript.

Figure 12.  Black text on purple background is nearly illegible.

We will change the colors to improve the readability of this figure in our revised manuscript.

Figure 14. For clarity, would suggest numbering the zones in the figure.

We will add the numbers of the zones in the revised Figure 14.

Lines 692-695. For clarity of discussion, would suggest using zone microphysical names rather than numbers here.

We will add the microphysical names of the zones in the discussion section.

---

## Author Response (AR2)

**Reviewer Report 1**

I appreciate the authors' hard work of addressing all comments from the reviewer. The revised manuscript has greatly improved its readability and has also addressed all my concerns. Therefore, I recommend this manuscript for publication.

We appreciate the valuable comments from the reviewer.
Our responses are in blue below each comment from the reviewer.

Minor comments:

(1) Line 664: it may not be suitable for...

Modified as suggested.

(2) Line 930: I wonder if this is the correct format for citing ATBDs, (the format is also inconsistent with the other ATBD you cited on Line 1005). Please check other's manuscripts for reference.

The Algorithm Theoretical Basis Document (ATBD) for Optimal Cloud Analysis Product is an online product guidance document from the Europe's Meteorological Satellite Agency (EUMETSAT). Therefore, we used this format in the reference.

(3) Also general for the reference section, please double-check that the format is consistent across all references, and make sure you have listed all references in the revised manuscript.

We double-checked the reference section, and all references were included.

**Reviewer Report 2**

The authors have addressed most of my concerns. Here are my remaining comments:

We appreciate the valuable comments from the reviewer.
Our responses are in blue below each comment from the reviewer.

(1) The T-ER figures show nominal ER values below (at larger T) the aircraft measured cloud base. These values are not physical and must combine radiance from the cloud and the surface. Please add a discussion that highlights this problem and the way it may affect the inference of the seedability. For example, it erroneously increases the indicated depth of the diffusional growth zone. There is such an example in Figure 9e.

We agree with the reviewer that we should be careful about the satellite data at a high temperature, which might be below the cloud base. For the case in Figure 9, the cloud base captured by the aircraft was about 9.1°C (Figure 6a). Meanwhile, according to the Skew-T chart (see figure below) from the observed sounding in Abu Dhabi at 00 UTC on the same day of this cloud case (closest sounding observation), the lifted condensation level (LCL, approximating the cloud base) was 15°C. That is warmer than the cloud base temperature captured by aircraft (~9.1°C), possibly due to the differences in time and locations between the sounding and aircraft observations. If the cloud base temperature from a sounding observation at the same time and location as the cloud case is available, it can be used to accurately exclude the satellite data below the cloud base and inhibit the corresponding uncertainties in the depth of the diffusional growth zone (Zone 1) if it is identified.

We replotted Figure 9 to exclude the satellite data below the cloud base (15°C) and clarified that in the caption. Meanwhile, we added the discussion above in the manuscript after the description of Figure 9.

[Figure]

*Figure: Skew-T chart from the sounding observation in Abu Dhabi at 00 UTC on August 18, 2019. (Credit: University of Wyoming)*

(2) Line 618-619: The text reads: "This is to identify the best targets for cloud seeding using hygroscopic particles, which is most effective in cases when coalescence is not active." But there are different kinds of hygroscopic seeding – small particles to suppress coalescence and invigorate the clouds or large particles to accelerate coalescence and make the cloud rain faster. You have to specify that the hygroscopic seeding that might work there is one that is aimed at enhancing drop coalescence and accelerating warm rain processes.

We have added two sentences for clarification (copied below) after the original sentence.

"When we reference hygroscopic seeding, we assume that the objective is to enhance precipitation. In this case, the hygroscopic particles are ultra giant cloud condensation nuclei (UGCCN), which when introduced at the cloud base and in the updraft region enhance the coalescence of warm-based clouds, accelerating the warm rain process (Rosenfeld et al., 2010)."

*Rosenfield, D., Axisa, D., Woodley, W. L., and Lahav, R.: A quest for effective hygroscopic cloud seeding, J. Appl. Meteorol. Climatol., 49, 1548-1562. doi:10.1175/2010JAMC2307.1, https://doi.org/10.1175/2010JAMC2307.1, 2010.*

(3) Line 630: Again, which kind of hygroscopic seeding do you refer to here?

The hygroscopic seeding here is similar to the hygroscopic seeding we mentioned in our response to the last comment. We modified the sentence in the manuscript to clarify that.

(4) Line 631: Zone 3 is defined here as the diffusional growth of supercooled cloud droplets. Supercooled clouds with drop coalescence are common but excluded from any zone's conditions. This gap has to be filled.

We agree with the reviewer that drop coalescence can occur in supercooled clouds, where collisions between supercooled droplets lead to the formation of larger, more massive droplets. A few studies reported observations of supercooled drizzle drops formed via coalescence processes when there were sufficient large droplets (Cober et al., 1996; Kajikawa et al., 2000). However, coalescence mainly applies to warm clouds where water droplets of many different sizes are swept upwards at different velocities so that they collide and combine with other droplets. The coalescence in supercooled clouds is typically rare and less efficient than in a warm cloud. That is because the precipitation-forming process in supercooled clouds is often dominated by the Bergeron-Findeisen process, when ice crystals form and rapidly grow by capturing supercooled water droplets.

In this study, the "supercooled water zone" (Zone 3) is defined as the zone of small, supercooled droplets with a slow droplet growth rate as shown in Figure 8. As we mentioned in the description of Zone 3 in Section 3.4, it is designed by the requirements of the cloud seeding operator to conduct hygroscopic and glaciogenic seeding. When Zone 3 exists and is sufficiently deep, the precipitation-forming processes are usually suppressed, indicating potential for rainfall enhancement by hygroscopic and/or glaciogenic seeding. The supercooled water droplets with coalescence may be partially covered by Zone 2 (droplet coalescence growth zone), in which the lower temperature threshold is -10°C and the droplet growth is quick. However, given the limited aircraft observation (so lack of validation), we are not confident to technically parameterize the occurrence of supercooled clouds with coalescence.

Because of the reasons discussed above and the goal of this study (categorizing microphysical zones to identify cloud seeding targets), the supercooled water with coalescence is not defined as a separate zone in this study.

Cober, S. G., Strapp, J. W., & Isaac, G. A. (1996). An example of supercooled drizzle drops formed through a collision-coalescence process. Journal of Applied Meteorology, 35(12), 2250-2260. https://doi.org/10.1175/1520-0450(1996)035<2250:AEOSDD>2.0.CO;2

Kajikawa, M., Kikuchi, K., Asuma, Y., Inoue, Y., & Sato, N. (2000). Supercooled drizzle formed by condensation–coalescence in the mid-winter season of the Canadian Arctic. Atmospheric research, 52(4), 293-301. https://doi.org/10.1016/S0169-8095(99)00035-6

(5) Line 649 and many other places: What are "suppressed microphysical processes"? I guess that you mean "suppressed precipitation-forming processes". Please replace all such occurrences with what is meant by this expression.

We agree with the reviewer that the term "precipitation-forming processes" is more accurate. We have changed suppressed/active "microphysical processes" to suppressed/active "precipitation-forming processes" throughout the manuscript, including the names in Figure 8 and Figure 14.

---

## Author Response (AR3)

**Reviewer Report 1**

We appreciate the valuable comments from the reviewer.
Our responses are in blue below each comment from the reviewer.

The authors claim that they have thoroughly checked the reference list. I did a quick scan of the main text, and the following citations were missing from the References List:

(1) Feng et al. 2018

We added Feng et al. 2018 in the Reference list.

(2) Fu et al. 2022

We added Fu et al. 2022 in the Reference list.

(3) Larzi et al. 2014 (Typo to Lazri et al. 2014)

We corrected this typo.

(4) Zipser et al 2006

We added Zipser et al 2006 in the Reference list.

In addition, we reviewed all citations in the manuscript and the reference list to ensure that each citation is included in the reference list.